# Benchmarking and Enhancing Large Language Models for Biological Pathway Reasoning

## Abstract

Large language models (LLMs) have demonstrated remarkable performance across various domains of biology, but their ability to reason about biological pathways remains underexplored. This includes reasoning about how perturbations in biological systems lead to various downstream effects through complex intermediate processes. Such reasoning is crucial for explaining and predicting biological phenomena, as well as for formulating hypotheses and designing experiments.

In this study, we investigate whether LLMs can effectively understand and reason about biological pathways by introducing BioMaze, a comprehensive benchmark focusing on reasoning about the effects and mechanisms of natural and synthetic interventions—such as mutations, infections, or treatments—on various downstream targets under different conditions through complex intermediate pathway processes. BioMaze spans multiple biological domains and is categorized along three reasoning dimensions, capturing various aspects of pathway reasoning.

We evaluate LLMs using the BioMaze benchmark with reasoning methods like Chain-of-Thought (CoT) and pathway graph-augmented approaches. Results show that while LLMs can understand mechanisms in natural organisms, they struggle with predicting phenomena after perturbations, highlighting their limitations in reasoning about biological pathways. To address these challenges, we propose PATHSEEKER, a novel LLM agent that interactively reasons through subgraph-based navigation within pathway graph. This approach enhances LLMs' reasoning in biological pathways by leveraging pathway graph augmentation, particularly in cases involving perturbations, potentially bridging the gap between LLMs' current capabilities and the complexities of biological systems.

## 1 Introduction

Large Language Models (LLMs) have recently shown impressive performance in science across various domains, including mathematics (Yu et al., 2023), chemistry (Liu et al., 2023b), biology (Hayes et al., 2024; Madani et al., 2020), and materials science (Zheng et al., 2023; Park et al., 2024). In the biological domain specifically, recent studies have demonstrated the potential of LLMs in tackling challenging tasks such as protein design (Valentini et al., 2023; Hosseini et al., 2024), drug discovery (M. Bran et al., 2024; Liu et al., 2023c), clinical trial analysis (Singhal et al., 2023; Jin et al., 2023), and experiment design (AI4Science & Quantum, 2023). Although LLMs are increasingly capable of addressing more complex, real-world problems within the biological sciences, their fundamental understanding, reasoning, and metacognitive abilities (Wei et al., 2022; Wang et al., 2022; Kojima et al., 2022) toward these scenarios—specifically in comprehending and reasoning through the intricate, multi-step processes involved in biological systems—have yet to be thoroughly explored.

Biological systems are composed of complex networks called pathways, which function as interconnected units involving various components, such as enzymes, substrates, and signaling molecules. These components interact in a highly coordinated manner, enabling the integration of multiple signals and precise regulation of system responses. As a result, intervention in a single component of a pathway—such as mutations, inhibitions, or pathogen infections—can influence other components within the organism via intricate, multi-step intermediate processes.

Figure 1: Illustration of BioMaze task and reasoning method with or without additional biological pathway graph data guidance. The task of BioMaze focuses on reasoning about the effects and mechanisms of natural components or synthetic interventions on various downstream targets under different conditions through complex intermediate pathway processes.

A wide range of biological phenomena can be explained and predicted by understanding and reasoning about biological pathways. In biological research, this understanding is essential for formulating hypotheses, designing experiments, and predicting and interpreting results. For example, blocking muscarinic M3 receptors in taste cells triggers a sequence of events, including reduced calcium mobilization in Type II taste cells, a weakened role of CALMH1 in ATP release, and diminished taste-evoked responses in taste sensory fibers, as shown in Figure 1. These insights can be useful for toxicity analysis, designing experiment groups with induced taste suppression, and developing treatments for decreased sense of taste.

Given the complexity of biological systems and the importance of understanding pathway interactions, the application of LLMs to analyze and predict their behavior presents both opportunities and challenges. In this paper, we first introduce BioMaze, a benchmark that serves as a crucial starting point for assessing LLMs' ability to comprehend and reason about realistic biological pathway phenomena. BioMaze compiles biological pathway phenomena from literature and generating corresponding questions and answers. These questions span multiple biological domains, focusing on predicting the effects and mechanisms of natural and synthetic interventions on various targets under different conditions through complex intermediate processes. Targets may include individual components, component interactions, their roles in biological processes, or larger-scale functions.

We conducted extensive evaluations of LLMs using the BioMaze benchmark, incorporating reasoning methods such as Chain-of-Thought (CoT) and pathway graph-augmented approaches (Li et al., 2023a; Sun et al., 2023; He et al., 2024). The results show that while LLMs demonstrate an understanding of mechanisms within natural organisms, they struggle to predict phenomena and grasp mechanisms when perturbations are introduced into the system—such as during interventions or when organisms are in altered conditions. This reveals that LLMs' causal reasoning abilities for biological pathways are limited. To address these challenges, we then propose a novel approach, PATHSEEKER, an LLM agent that interactively reasons through subgraph-based navigation while exploring the pathway graph. This method enhances LLMs' performance in complex biological reasoning tasks by effectively leveraging pathway graph information as blueprints in reasoning, especially in the case of interventions. In summary, our contributions are as follows:

- We introduce BioMaze, a comprehensive benchmark designed to assess LLMs' ability to understand and reason about biological pathways. BioMaze focuses on evaluating the models' capacity to predict the effects and elucidate the mechanisms of both natural and synthetic interventions—such as mutations and infections—on various downstream targets under diverse conditions through complex intermediate pathway processes. The benchmark spans multiple biological domains and is structured along three dimensions: interventions, conditions, and target types.

- We conduct extensive evaluations of LLMs using BioMaze, incorporating advanced reasoning methods such as CoT and pathway graph-augmented approaches. Our results reveal that while LLMs demonstrate proficiency in understanding mechanisms within natural organisms, they encounter significant challenges when predicting phenomena and comprehending mechanisms in perturbed systems. These findings highlight critical limitations in LLMs' reasoning capabilities within the domain of biological pathways.

- We propose PATHSEEKER, a novel LLM agent approach that employs interactive, subgraph-based exploration to navigate pathway databases during reasoning. This method enhances LLMs' reasoning in biological pathways by leveraging pathway graphs as structured blueprints, especially for the case with interventions, potentially bridging the gap between LLMs' current capabilities and the complexities of biological systems.

## 2 RELATED WORK

**Biological Scientific Question Answering** Previous studies have explored the potential of language models in the biological scientific domain (Lu et al., 2022; Vilares & Gómez-Rodríguez, 2019; Jin et al., 2021; Pal et al., 2022). MEDHOP (Welbl et al., 2018) and PubMedQA (Jin et al., 2019) investigated biological scientific QA in the form of reading comprehension. BioASQ-QA (Krithara et al., 2023) proposed a realistic question-answering benchmark for the information needs of biomedical experts. A few studies have examined language models' ability to understand biological pathways. Chatpathway (Li et al., 2023b), Park et al. (2023), and Azam et al. (2024) specifically investigated language models' capacity for completing biological pathways. Different from previous tasks, this work introduces a novel task for practical biological pathway phenomenon reasoning, addressing the reasoning capacity of LLMs. See Appendix A.13 for a more detailed comparison.

**Graph-augmented Language Model** Several studies have explored augmenting LLMs with graph data. In particular, some works enhance LLMs by encoding graph data as text (Ye et al., 2023; Wang et al., 2024; Fatemi et al., 2023), or tuning LLMs specifically for graph-based tasks (Liu et al., 2023a; Tang et al., 2024; He et al., 2024; Zhao et al., 2023; He & Hooi, 2024). Augmented LLMs have been applied to knowledge-based QA (Sun et al., 2023; He et al., 2024; Li et al., 2023a; Jin et al., 2024; Cheng et al., 2024; Edge et al., 2024) and graph tasks like property prediction (Wang et al., 2024; He et al., 2023). Some studies leverage graph structures for complex reasoning tasks (Jiang et al., 2023; Besta et al., 2024). Most large graph databases use retrieval mechanisms (He et al., 2024; Li et al., 2023a), while others employ LLMs as interactive agents for navigation (Sun et al., 2023; Jin et al., 2024; Li et al., 2024). This work introduces a more efficient agent-based approach using subgraph navigation and reasoning to improve pathway database exploration.

## 3 BENCHMARK: BIOMAZE

### 3.1 DATASET CREATION

To gather realistic biological pathway phenomena, particularly those involving interventions, as illustrated in Figure 1, the data for BioMaze is sourced from over 6,000 biological pathway research papers. These studies involve carefully designed intervention experiments supported by pathway mechanisms to observe how biological systems respond. We extract detailed experimental observations and their contexts directly from the abstracts. Importantly, our focus is on the specific experimental phenomena observed and reported, rather than the final conclusions drawn by the researchers. This is essential for our goal of predicting detailed events in the benchmark. The dataset creation involves prompting the large language model, and in this study, we choose LLaMA3.1-405B (Dubey et al., 2024) and GPT-4 as the models for data creation.

After extracting the phenomena, we convert each one into either a True/False or open-ended question, depending on its content. Each question is paired with corresponding labeled answers. We then apply multiple data filters and human validation steps to ensure the correctness, quality, and relevance to biological pathways. The correctness of each question is validated by checking whether LLMs can answer it accurately using the original paper content. Question quality is ensured through several filters, removing questions that are poorly defined, unpredictable (e.g., asking for specific measurement values), query more than one fact, are trivial with answers revealed in the question's context, or are unrelated to biological pathways. After all the filters, BioMaze contains 1.3k high-quality questions for biological pathways reasoning.

The questions of BioMaze cover a wide range of biological domains, including metabolism, genetic information processing, environmental information processing, cellular processes, organismal systems, and human diseases. The biological domain distribution is illustrated in Figure 2 (left).

Table 1: Task example for each category.

| Dimension | Category | Example (abbreviated) | Illustration |
|---|---|---|---|
| Inquiry Type | Normal | What is the effect of AMPK activation on SIRT1 activity in mouse skeletal muscle? | |
| | Perturbed | What is the effect of GogB-deficient Salmonella on NFkappaB activation and proinflammatory responses in infected mice? | |
| Extra Condition | Natural | How does apelin affect TNFalpha inhibition on brown adipogenesis? | |
| | Intervened | What is the role of BID in BAX activation in AIF-mediated necroptosis after MNNG treatment? | |
| Investigation Target | Single | What happens to AQP2 upon ADH stimulation? | |
| | Interaction | How does the influenza protein NS1 affect the activation of RIG-I by viral ssRNA? | |
| | Function | What is the effect of losing 11beta-HSD2 from the fetus and fetally derived tissues on cerebellum development? | |

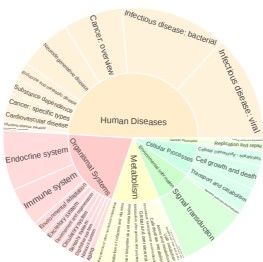 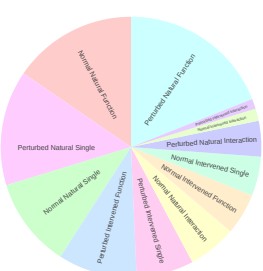

Figure 2: Dataset biological domain and reasoning type distribution. Left: BioMaze covers six main domains: metabolism, genetic information processing, environmental information processing, cellular processes, organismal systems, and human diseases. Right: BioMaze is categorized along three dimensions of reasoning types: inquiry type, extra condition, and investigation target.

## 3.2 REASONING TYPE CATEGORIES

The questions in BioMaze are categorized across three dimensions: inquiry type, extra condition, and investigation target, leading to varying types and difficulties of reasoning, as shown in Table 1. More full question cases are in Appendix A.1. The distribution of the three dimensions' questions is shown in Figure 2 (right). We introduce each category of the dimensions below:

**Dimension 1: Inquiry Type**

**Category 1: Normal Source** This category involves predicting the effects of natural components in their normal state within a biological pathway. Tasks here focus on understanding the fundamental mechanisms of pathways in biological systems. The goal is to evaluate how well LLMs can comprehend and explain typical biological pathway functions.

**Category 2: Perturbed Source** This category deals with predicting the effects of external interventions or treatments, such as mutations, infections, or experimentally introduced elements, on downstream targets within pathways. Tasks emphasize reasoning about how these interventions alter pathway functions. This mirrors real-world biological research, where the focus is often on understanding how such interventions influence biological systems and their downstream targets.

**Dimension 2: Extra Condition**

**Category 1: Natural Condition** In this category, the task predicts the impact of the inquiry source under an organism's natural conditions, meaning no additional treatments are applied to the biological system beyond the inquiry source. For example, in Table 1, the natural condition question asks about the mechanism through which apelin affects TNF-alpha inhibition in brown adipogenesis, with no extra interventions present in the pathway.

**Category 2: Intervened Condition** This category assesses the inquiry source's impact when combined with other factors like mutations, infections, or interventions, focusing on how these conditions alter the pathway. For example, the question in Table 1 examines BID's role after MNNG treatment, where the pathway differs from its natural state. Enhancing performance here is crucial for modeling complex biological scenarios, such as predicting treatment outcomes and drug interactions, as it shows how multiple factors interact within a system.

**Dimension 3: Investigation Target**

**Category 1: Single Component as Target** This category focuses on investigating the effect of the source on a specific component within the pathway, such as its expression, activation, or inhibition.

**Category 2: Components Interaction as Target** This category examines the effect of the source on interactions between components within the pathway. It may involve understanding how downstream components interact with each other or their roles in regulating pathway processes. For example, the question of this category in Table 1 queries influenza protein NS1's effect on the downstream process that viral ssRNA activates RIG-I.

**Category 3: Function as Target** This category evaluates the effect of the source on broader biological functions or macro-level phenomena within the organism. It addresses more comprehensive system behaviors, helping to link pathway-level changes with organism-wide outcomes, which are crucial for scenarios like understanding health and disease processes.

### 3.3 PATHWAY GRAPH AUGMENTATION

Text-only reasoning methods like CoT generate reasoning steps directly from LLMs based on a given question. However, reasoning about biological pathways poses challenges due to their graph-based nature, requiring LLMs to not only have an implicit understanding of these pathways but also effectively plan and execute complex reasoning steps.

This work addresses the question: Do large language models require pathway graph data augmentation to reason effectively about biological systems? Providing explicit pathway graphs could serve as structural blueprints, enhancing reasoning from both knowledge and logic perspectives. We formalize this problem as:

$$a = G(\mathcal{E}, o), \tag{1}$$

where $G$ represents the language model, $\mathcal{E}$ denotes the task instruction (including the question), $o$ refers to the observation from the augment pathway graph database, and $a$ signifies the model output which could be the answer as well as the reasoning process.

**Pathway Graph Database** To augment LLMs with reasoning in biological pathways, we created a pathway graph database based on KEGG (Kanehisa & Goto, 2000), a collection of pathway maps on metabolism and various cellular and organismal functions widely-used resource among biologists. We compiled all available pathway networks and maps from KEGG and integrated all of them into a single pathway graph database. The statistics for the pathways are in Appendix A.5. Each entry in the dataset is provided with a detailed description and function corpus. The graph is structured in triples as [Head IDs, Tail IDs, (Relation Type, Biological Process IDs)].

**Pathway Graph Database API** When the language model accesses the pathway database, it may need to retrieve relevant triples from the pathway graph using APIs like Search_Node, Search_Edge, and Search_Triple (Sun et al., 2023; Li et al., 2023a). Our pathway database supports these core retrieval APIs based on detailed descriptions and functional corpora. These APIs are essential for enabling various graph-augmented reasoning methods in LLMs.

Since the connectivity of the pathway graph is crucial for enhancing reasoning in biological systems, we also developed the retrieval API designed to find the optimal connected subgraph $S^* = \text{Search\_Subgraph}(query, N)$, where $S^*$ is the retrieved subgraph, $query$ represents the query content, such as keywords, and $N$ is the target size. The goal is to match a given target size as closely as possible while maximizing the matching score. This is formulated as a optimization problem:

$$S^* = \text{Search\_Subgraph}(query, N) = \underset{S \subseteq P, S \text{ is connected}, |S|=N}{\text{argmax}} \sum_{i \in V_S \cup E_S} \text{score}(i, query) \tag{2}$$

Here, the overall pathway network is denoted by $P$, with $V_S$ and $E_S$ representing the node and edge sets of subgraph $S$. The details of the implementation are described in Appendix A.6.

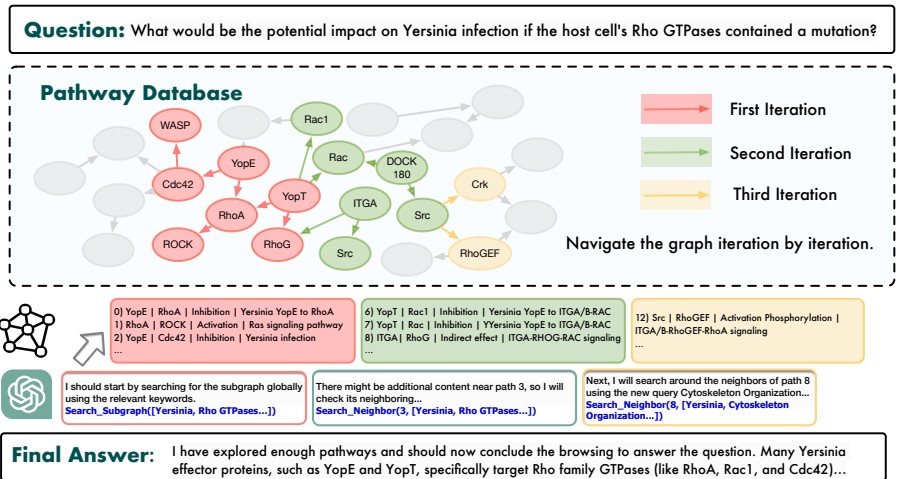

Figure 3: PATHSEEKER allows interactive browsing of the pathway graph database by navigating through subgraphs. At each step, PATHSEEKER can perform either a global subgraph search or a local search around a previously explored pathway step. This functionality enables PATHSEEKER to fully leverage the augmented pathway graph database during biological pathway reasoning.

**Graph-to-Text Encoder** For retrieved results which is a list of triples (e.g., the output of Search_Triple or Search_Subgraph), the result $S$ is encoded into text $o$ using the following process:

$$o = \text{TripleToText}(\text{DFSOrder}(S)) \tag{3}$$

Function DFSOrder arranges the triples in depth-first search (DFS) order. Unlike other methods, such as relevance scoring, DFS order more closely aligns with the reasoning process through the subgraph, reflecting the natural progression of biological functions. The ordered subgraph $\hat{S}$ is converted into text format for the LLM by encoding each triple as a string in the following format: Head | Tail | Relation and Biological Process.

# 4 METHOD: PATHWAY REASONING AGENT PATHSEEKER

As we evaluated several graph-augmented reasoning methods, we found that current graph-augmentation methods' performance is limited by their ineffective utilization of the pathway graph database for reasoning. In this work, we propose a general solution for biological pathway reasoning called PATHSEEKER. This solution takes the form of a reasoning agent that can interactively perceive and navigate pathways using a web-like engine, along with flexible reasoning in each step.

**Subgraph Navigation-based Graph Browsing** Inspired by how humans browse web networks, PATHSEEKER allows the language agent to flexibly explore a vast graph database by observing subgraphs at each step, as shown in Figure 3. At step $t$, the language agent $G$ takes an action step $a_t$ based on problem $\mathcal{E}$ (problem instructions) and previous observation-action trajectory $h_t = [o_1, a_1, \ldots, o_{t-1}, a_{t-1}, o_t]$,

$$a_t = G(\mathcal{E}, h_t) \tag{4}$$

In addition to the global subgraph retriever Search_Subgraph, PATHSEEKER has access to an additional neighbor subgraph retriever, Neighbor_Subgraph($line\_id, query, N$), which retrieves an optimal connected subgraph of target size from the multi-hop neighbors of a previously observed pathway step $line\_id$.

$$\text{Neighbor\_Subgraph}(line\_id, query, N) = \underset{S \subseteq P_{id}, S \text{ is connected}, |S|=N}{\operatorname{argmax}} \sum_{i \in V_S \cup E_S} \text{score}(i, query) \tag{5}$$

Here, $P_{id}$ represents the multi-hop neighbors of the triple with $line\_id$. This allows PATHSEEKER to navigate the pathway graph database by either performing a global search or by exploring the multi-hop neighbors of an observed subgraph at each step. See Appendix for case A.7.

**Graph Encoding** In step $t$, the action taken by LLM agent get subgraph $S_t$ from environment, and the subgraph is encoded into text observation $o_t$ as following:

$$\hat{S}_t = \text{DFSOrder}(\text{RemoveSeen}(S_t, [S_1, \ldots, S_{t-1}]))$$
$$o_t = \text{TripleToOrderedText}(\hat{S}_t, \text{TotalNum}([S_1, \ldots, S_{t-1}]))$$
(6)

Function $\text{RemoveSeen}$ eliminates triples from the $t$-th turn's subgraph that have been observed in previous turns, ensuring that each triple appears in the LLM's observations only once when first retrieved. This approach enhances content length efficiency and encourages the LLM to understand the whole navigation history rather than focusing solely on the most recent turn.

The function $\text{TripleToOrderedText}$ convert ordered subgraph $\hat{S}_t$ into text in the following format: Line ID) Head | Tail | Relation and Biological Process. These global line IDs indicate the order of each triple across all turns, providing a unique reference for the LLM agent during local searches or reasoning. For the $t$-th turn's subgraph $S_t$, the ID starts at the total number of unique triples seen in previous history, given by $\text{TotalNum}([S_1, \ldots, S_{t-1}])$.

**Final Reasoning** As graph data browsing finishes, the final reasoning is conducted based on all the navigation history:

$$a_r = G(\mathcal{E}_r, [o_1, \ldots, o_T])$$

**Graph Navigation Capacity** The combination of global and local subgraph retrieval APIs empowers LLM agents to explore the entire network flexibly and efficiently. It allows the LLM to guide its exploration by adjusting both keywords and the root of the local subgraph, depending on the intermediate reasoning, offering stronger expressiveness than navigation methods like BFS, DFS, and various retrieval methods.

## 5 EXPERIMENT

### 5.1 BASELINE AND METRIC

We evaluate the reasoning performance of LLMs on BioMaze in both the unaugmented step-by-step reasoning and the pathway graph-augmented methods. We adopt reasoning method without graph augmentation Chain-of-Thought (CoT) (Wei et al., 2022; Kojima et al., 2022), and methods with pathway graph augmentation: Chain-of-Knowledge (CoK) (Li et al., 2023a), Think-of-Graph (ToG) (Sun et al., 2023), and G-Retriever (He et al., 2024). Details of baselines are in Appendix A.8.

For True/False tasks, we compute accuracy averaged across the True and False labels to account for label imbalance in the dataset. For open-ended tasks, the LLM is used to evaluate the accuracy of generated answers by comparing them to the ground truth and determining whether they are correct or incorrect. In this study, we use the LLaMA3.1-405B model as the evaluator, with five in-context examples. The performance of the evaluator is further analyzed in Appendix A.9.

### 5.2 MAIN RESULT

We evaluate PATHSEEKER and baseline methods on BioMaze, presenting results in Tables 2 and 3. The comparison covers task dimensions including signal source, additional conditions, and target. The results lead to the following conclusions:

**LLMs can perform biological system reasoning tasks.** Despite the extreme difficulty of the tasks in BioMaze, LLMs still achieved strong results, especially in normal inquiry and natural condition cases. The overall performance suggests that LLMs are capable of reasoning in many biological research scenarios, effectively explaining and predicting phenomena within biological systems.

**Question with perturbation query in BioMaze presents significant reasoning challenges for LLMs.** Specifically, LLMs struggle more with perturbed inquiry type settings than with normal inquiry types in both True/False and open-ended formats. This suggests that reasoning about biological pathways becomes more complex in intervention scenarios, where the events are less likely to align with common biological knowledge and require deductive reasoning to predict pathway behavior. In contrast, questions set in normal scenarios are more likely to be answerable using established biological knowledge about how typical pathways work.

Table 2: Accuracy (%) on BioMaze True/False tasks (50% corresponds to the random guessing baseline). For each method, the lowest result within each dimension is underlined to highlight the most challenging setting.

| | w.t. Pathway Graph | Inquiry Type | | Extra Condition | | Investigation Target | | |
| --- | --- | --- | --- | --- | --- | --- | --- | --- |
| | | Normal | Perturbed | Natural | Intervened | Single | Interaction | Function |
| GPT-3.5 | | | | | | | | |
| Viliana (0 Shot) | | 76.80 | 67.42 | 74.30 | 66.23 | 68.90 | 78.97 | 71.44 |
| Viliana (2 Shot) | ✗ | 72.09 | 70.22 | 71.28 | 70.28 | 70.48 | 81.24 | 67.15 |
| CoT (0 Shot) | | 78.02 | 65.45 | 75.08 | 64.35 | 68.45 | 69.75 | 75.23 |
| CoT (2 Shot) | | 77.03 | 67.13 | 73.65 | 68.92 | 68.92 | 79.26 | 71.85 |
| ToG | | 74.57 | 69.66 | 74.17 | 68.04 | 70.03 | 73.67 | 73.80 |
| CoK | ✓ | 77.47 | 68.54 | 73.09 | 72.95 | 67.92 | 80.56 | 73.86 |
| G-Retriever | | 76.26 | 70.20 | 75.72 | 70.81 | 73.04 | 76.21 | 73.59 |
| PATHSEEKER | | 78.85 | 74.44 | 77.63 | 74.36 | 78.01 | 81.66 | 73.78 |
| LLaMA3 8B | | | | | | | | |
| Viliana (0 Shot) | | 80.49 | 67.70 | 76.78 | 67.17 | 75.27 | 73.88 | 73.27 |
| Viliana (2 Shot) | ✗ | 80.19 | 72.75 | 78.42 | 70.69 | 79.72 | 83.18 | 70.85 |
| CoT (0 Shot) | | 75.07 | 67.13 | 72.04 | 68.66 | 73.15 | 80.15 | 66.33 |
| CoT (2 Shot) | | 81.77 | 71.63 | 79.04 | 70.67 | 79.73 | 84.35 | 71.52 |
| ToG | | 79.37 | 69.31 | 76.96 | 67.60 | 76.57 | 83.17 | 69.16 |
| CoK | ✓ | 80.20 | 67.70 | 75.87 | 69.00 | 77.27 | 81.11 | 68.93 |
| G-Retriever | | 80.59 | 72.29 | 81.17 | 70.06 | 80.97 | 82.29 | 73.53 |
| PATHSEEKER | | 83.08 | 75.84 | 82.14 | 72.27 | 81.07 | 86.62 | 75.01 |

Table 3: Accuracy (%, evaluated by LLM) on BioMaze open-ended tasks. For each method, the lowest result within each dimension is underlined to highlight the most challenging setting.

| | w.t. Pathway Graph | Inquiry Type | | Extra Condition | | Investigation Target | | |
| --- | --- | --- | --- | --- | --- | --- | --- | --- |
| | | Normal | Perturbed | Natural | Intervened | Single | Interaction | Function |
| GPT-3.5 | | | | | | | | |
| CoT (0 Shot) | ✗ | 76.60 | 67.67 | 72.93 | 68.28 | 73.20 | 64.86 | 71.50 |
| CoT (2 Shot) | | 82.67 | 73.66 | 79.66 | 72.69 | 83.28 | 63.51 | 75.73 |
| ToG | | 74.77 | 65.27 | 70.81 | 66.08 | 72.13 | 62.16 | 68.60 |
| CoK | ✓ | 82.98 | 73.43 | 80.41 | 70.93 | 82.30 | 67.57 | 75.73 |
| G-Retriever | | 84.38 | 72.84 | 80.78 | 74.37 | 82.55 | 70.40 | 76.92 |
| PATHSEEKER | | 87.84 | 77.91 | 83.65 | 78.85 | 85.29 | 77.03 | 80.74 |
| LLaMA3 8B | | | | | | | | |
| CoT (0 Shot) | ✗ | 82.37 | 69.53 | 77.63 | 69.16 | 76.14 | 62.16 | 76.78 |
| CoT (2 Shot) | | 80.55 | 67.91 | 75.94 | 67.40 | 77.78 | 55.41 | 73.35 |
| ToG | | 84.80 | 73.49 | 80.64 | 73.13 | 82.68 | 74.32 | 75.73 |
| CoK | ✓ | 80.55 | 70.70 | 77.82 | 68.28 | 78.43 | 64.86 | 74.14 |
| G-Retriever | | 82.62 | 72.10 | 77.21 | 75.92 | 80.71 | 72.53 | 75.65 |
| PATHSEEKER | | 84.50 | 76.51 | 80.64 | 78.41 | 83.01 | 78.38 | 77.84 |

**Questions with intervened conditions present greater reasoning challenges.** Similar to the inquiry type, interventions in the extra condition also create difficulties for LLMs in reasoning. These interventions, such as disruptions caused by external factors like mutations, infections, or experimental setups, complicate the biological system's mechanism. As a result, reasoning in these scenarios relies less on established knowledge of natural biological systems and more on deductive reasoning to navigate the altered conditions.

**Reasoning target brings diverse challenges for reasoning.** The Investigation target presents varied difficulties, resulting in inconsistent performance across different backbone models and reasoning methods. Interestingly, the performance of True/False questions varies between GPT-3.5 and LLaMA3, likely due to differences in the knowledge underlying the two models. Additionally, open-ended questions about interactions are the most challenging, which differs from the True/False format. This may be due to the nature of open-ended questions, which have a broader possible answer space for questions about interaction.

**Pathway augmentation can enhance reasoning in biological systems, especially for intervention cases.** As shown in Tables 2 and 3, reasoning methods with pathway augmentation, especially PATHSEEKER, outperform non-augmented approaches. PATHSEEKER consistently exceeds CoT across all question types and categories, regardless of the backbone model, highlighting the value of integrating biological pathways to enhance reasoning in biological systems. Additionally, PATHSEEKER outperforms other graph augmentation methods, demonstrating the effectiveness of its subgraph-based navigation approach. Notably, it reduces the performance gap between natural and intervened/perturbed groups, helping bridge the gap in causal reasoning for biological pathways.

## 5.3 ANALYSIS

### 5.3.1 TASK ANALYSIS

**Reasoning Difficulty with Steps** While LLMs excel at complex reasoning by breaking down questions into basic steps for detailed deductive reasoning, it's widely recognized that the complexity of reasoning is closely tied to the foundational steps needed for a task. To explore the relationship between task complexity and reasoning steps in BioMaze, we analyzed the reasoning steps for each instance by prompting the LLaMA3.1-405B to explain their reasoning process based on the correct answer and pathway information.

Figure 4 shows the performance of Chain-of-Thought (CoT) reasoning. As the number of reasoning steps increases, CoT performance steadily declines, indicating that reasoning difficulty rises with a greater number of pathway steps. This finding supports our hypothesis that the challenges in biological pathway reasoning are largely due to the complexity of the pathways involved.

Notably, PATHSEEKER's performance remains more consistent across different reasoning step counts. This suggests that augmenting LLMs with biological pathway information can mitigate the challenges of pathway reasoning, particularly when dealing with intricate intermediate processes.

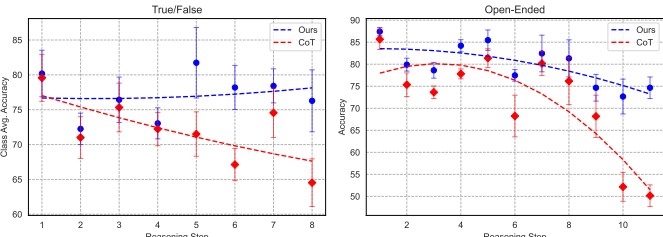

Figure 4: Performance versus reasoning steps. LLMs face increasing difficulty in reasoning about biological systems as task complexity rises and requires more reasoning steps. In contrast, pathway augmentation significantly mitigates the drop of performance for tasks that involve more steps.

**Reasons for Failure Statistics** To illustrate the reasons why LLMs fail in biological pathway reasoning, we analyze failed cases across various reasoning methods, including CoT and PATHSEEKER. The failure reasons are classified into the following categories: (1) **Unresolved Conclusion** For cases where the model fails to provide a definitive answer, indicating uncertainty or belief that the answer is unknown. (2) **Incomplete Answer** When the response lacks essential details, such as missing the requested effects or other key elements. (3) **Omission in Reasoning** For errors where critical pathway steps in the question's biological process are left out, causing the final answer to be incorrect. (4) **Faulty in Reasoning** When the reasoning path is correct, but there are significant errors in deducing the events within that pathway. We manually classify 100 random samples from these error cases to approximate the overall error cases, with a professional biology Ph.D. student.

The results are shown in Figure 5. The results in Figure 5 show that in both True/False and open-ended tasks, the main error in CoT reasoning is faulty reasoning, where LLMs correctly identify the biological pathway but misinterpret the events within it. Another key error is omission, where critical steps or branches of the pathway are overlooked. This highlights the challenges LLMs face in reasoning about biological pathways, due to both knowledge gaps and difficulties in deductive reasoning. PATHSEEKER significantly reduces faulty reasoning by providing pathway graphs, enabling more accurate reasoning about biological events. However, omissions remain a predominant issue, often due to limitations in the pathway database and oversights during browsing. Additionally, with the availability of pathways, LLMs are less likely to fail in providing definitive answers, becoming more confident in drawing conclusions.

**Performance with Biological Domain** Figure 6 presents a comparison of the performance of various reasoning methods across different biological domains in BioMaze. The results demonstrate that the difficulty of each domain varies depending on the reasoning method used. Overall, PATHSEEKER, when augmented with pathway information, consistently outperforms direct reasoning across nearly all biological domains. The results of more backbones are in Appendix A.11.

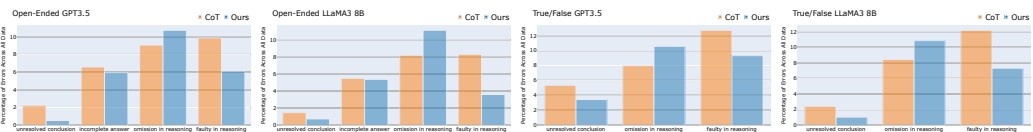

Figure 5: Error analysis for CoT reasoning and reasoning with pathway augmentation (our method PATHSEEKER). The primary cause of errors in (CoT) reasoning for biological systems is due to both faulty reasoning and omissions in reasoning. When pathway augmentation is applied, omissions in reasoning become the predominant issue, but the rate of faulty reasoning is significantly reduced, thereby improving the overall reasoning accuracy of LLMs in biological systems.

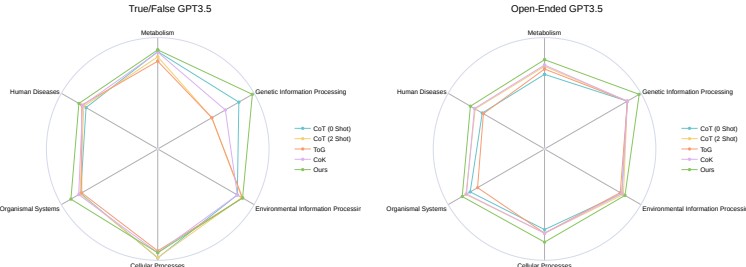

Figure 6: GPT-3.5's performance across different biological domains in BioMaze.

## 5.4 METHOD ANALYSIS AND ABLATION STUDY

**API Usage and Step Distribution** We analyze PATHSEEKER's agent behavior by reviewing task steps and navigation API usage frequency. Tables 4 and 5 show that over half of the tasks are completed in six steps or fewer, while some take over ten steps which is possible due to lacking relevant pathway data. On average, the agent performs 1.5 global searches per task, suggesting multiple searches are often needed, and local navigation occurs more than three times per task, highlighting thorough subgraph exploration.

Table 4: Agent steps distribution (%) of PATH-SEEKER during task completion.

| Agent Steps | 1-4 | 4-6 | 6-8 | 8-10 | ≥10 |
|---|---|---|---|---|---|
| True/False | 14.35 | 42.26 | 20.97 | 9.68 | 12.74 |
| Open-Ended | 19.50 | 41.77 | 15.94 | 8.43 | 14.36 |

Table 5: Average API usage times of PATH-SEEKER during task completion.

| | Global | Local |
|---|---|---|
| True/False | 1.51 | 3.40 |
| Open-Ended | 1.62 | 3.26 |

**Ablation Study** To assess the effectiveness of PATHSEEKER's components, we conduct ablation studies, with results for LLaMA3-8B shown in Table 6. The most impactful component is Final-Reaser; without it, the agent's answers suffer due to the long task history, disrupting reasoning and responses. The local search API is also critical, enabling efficient graph navigation. Lastly, the graph encoding method boosts performance, emphasizing the importance of encoding graph data for sequential language models.

Table 6: Ablation Study of PATHSEEKER.

| | PATHSEEKER | w.o. RemoveSeen | w.o. DFSOrder | w.o. Local search | w.o. FinalReasoner |
|---|---|---|---|---|---|
| True/False | 79.24 | 76.57 | 77.4 | 77.29 | 75.33 |
| Open-Ended | 79.97 | 77.52 | 77.02 | 76.27 | 71.86 |

## 6 CONCLUSION

In this study, we introduce BioMaze, a benchmark designed to evaluate LLMs' ability to understand and reason about biological pathways by predicting the effects of natural and synthetic interventions, like mutations and infections, on downstream targets. Extensive evaluations using BioMaze, incorporating advanced methods like CoT and pathway graph-augmented approaches, show that LLMs struggle with understanding pathway mechanisms with intervention. We also propose PATH-SEEKER, a novel LLM agent that uses interactive subgraph exploration to enhance reasoning in biological pathways by leveraging pathway graphs as structured blueprints.

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

# A APPENDIX

## A.1 DATASET CASE ILLUSTRATION

**Dimension 1: Inquiry Type**

**Category 1: Normal Source**

"Question": "AMP-activated protein kinase (AMPK) is a metabolic fuel gauge that senses changes in the intracellular AMP/ATP ratio. Recent evidence suggests that AMPK plays a role in the therapeutic benefits of metformin, thiazolidinediones, and exercise in the management of type 2 diabetes and associated metabolic disorders. AMPK controls the expression of genes involved in energy metabolism in mouse skeletal muscle by working together with another metabolic sensor, the NAD+-dependent type III deacetylase SIRT1. Does AMPK enhance SIRT1 activity by decreasing cellular NAD+ levels?",

"Answer": "No"

"Question": "Adrenergic receptor signaling in adipocytes controls the hydrolysis of triglycerides and is involved in brown adipocyte thermogenesis and energy consumption. Do beta-adrenergic receptors activate a network of signaling pathways that include cAMP-dependent protein kinase and members of the mitogen-activated protein kinase family?",

"Answer": "Yes"

"Question": "Enteropathogenic Escherichia coli (EPEC) is a human pathogen that colonizes the gut mucosa and causes diarrheal diseases. EPEC uses a type III secretion system (T3SS) to deliver effectors into host cells, which repress innate immune responses and infiltration of immune cells. One of these effectors is NleF. What is the effect of NleF on caspase-4 activity in intestinal epithelial cells (IECs) infected with EPEC?",

"Answer": "NleF inhibits the proteolytic activity of caspase-4 in IECs infected with EPEC."

"Question": "Epithelial-mesenchymal transition (EMT) of tubular epithelial cells is a key event in renal interstitial fibrosis and the progression of chronic kidney disease (CKD). Apelin is a regulatory peptide involved in the regulation of normal renal hemodynamics and tubular functions. To examine the effects of apelin on transforming growth factor-beta1 (TGF-beta1)-induced EMT in HK-2 cells, cells were co-treated with apelin and TGF-beta1. What is the effect of apelin on TGF-beta1-mediated upregulation of alpha-smooth muscle actin (alpha-SMA) and downregulation of E-cadherin in HK-2 cells?",

"Answer": "Apelin inhibits TGF-beta1-mediated upregulation of alpha-smooth muscle actin (alpha-SMA) and downregulation of E-cadherin in HK-2 cells."

**Category 2: Perturbed Source**

"Question": "To investigate the role of Dectin-1 in the innate response to mycobacteria, an in vitro system was used to measure IL-12p40 production in splenic dendritic cells (SpDC) exposed to live Mycobacterium tuberculosis bacilli. Does pharmacologic inhibition of spleen tyrosine kinase (Syk) reduce the IL-12p40 response induced by M. tuberculosis?",

"Answer": "Yes"

"Question": "Deficiency of GDP-Man:Man1GlcNAc2-PP-dolichol mannosyltransferase (hALG2) causes a new type of congenital disorder of glycosylation (CDG) called CDG-Ii. A patient with CDG-Ii showed symptoms such as mental retardation, seizures, coloboma of the iris, hypomyelination, hepatomegaly, and coagulation abnormalities. Skin fibroblasts from the patient exhibited an accumulation of Man1GlcNAc2-PP-dolichol and Man2GlcNAc2-PP-dolichol. The patient's fibroblast extracts were incubated with Man1GlcNAc2-PP-dolichol and GDP-mannose, and it was found that the mannosyltransferase activity elongating Man1GlcNAc2-PP-dolichol was severely reduced. The yeast ALG2 sequence was used to identify the human ortholog, and genetic analysis revealed that the patient had a single nucleotide deletion and a single nucleotide substitution in the human ALG2 gene. Was the expression of mutant hALG2 cDNA able to restore the mannosyltransferase activity and the biosynthesis of dolichol-linked oligosaccharides in both patient fibroblasts and yeast cells?",

"Answer": "No"

"Question": "Advanced glycation end products (AGEs) play a significant role in diabetic complications by activating various signaling pathways. One of the key pathways involved is the transforming growth factor (TGF)-beta signaling pathway, which regulates Smad proteins. To understand the role of Smad signaling in diabetic complications, researchers have investigated the effect of AGEs on Smad activation and collagen synthesis. What is the effect of overexpressing Smad7 on AGE-induced Smad activation and collagen synthesis?",

"Answer": "Overexpression of Smad7 prevents AGE-induced Smad activation and collagen synthesis."

"Question": "Aggregation of amyloid-beta (Abeta) and Tau protein are hallmarks of Alzheimer's disease (AD). According to the Abeta-cascade hypothesis, Abeta is considered toxic for neurons and Tau is a downstream target of Abeta. In differentiated primary hippocampal neurons, the effect of exposure to Abeta oligomers on the phosphorylation of Tau in dendritic regions was investigated. What is the effect of exposure to Abeta oligomers on the phosphorylation of Tau in dendritic regions?",

"Answer": "Exposure to Abeta oligomers leads to elevated phosphorylation of Tau at certain sites diagnostic of AD-Tau in dendritic regions."

**Dimension 2: Extra Condition**

**Category 1: Natural Condition**

"Question": "The expression and function of ENaC and Na,K-ATPase on the cell surface are tightly controlled by a complex regulatory network. Does aldosterone acutely regulate the expression of elements in this regulatory network that control the cell-surface localization and function of ENaC and Na,K-ATPase?",

"Answer": "Yes"

"Question": "Alveolar macrophages (AM) play a central role in initiating and resolving lung inflammation, but the integration of these opposing functions is not well understood. Cholesterol 25-hydroxylase (CH25H) is highly expressed in AMs and is responsible for the production of 25-hydroxycholesterol (25HC), which activates the anti-inflammatory nuclear receptor liver X receptor (LXR). Is CH25H required for LXR-dependent promotion of AM lipid overload?",

"Answer": "No"

"Question": "To investigate the role of CSS3 in CS production, researchers overexpressed CSS3 in HeLa cells and measured the resulting CS levels. Does overexpressing CSS3 increase the amount of CS in HeLa cells?",

"Answer": "Overexpressing CSS3 increases the amount of CS in HeLa cells."

"Question": "Chronic activation of Wnt/beta-catenin signaling is found in various human malignancies, including melanoma, colorectal, and hepatocellular carcinomas. What is the effect of HCMV infection on beta-catenin stabilization and signaling in cells?",

"Answer": "HCMV infection significantly increases beta-catenin stabilization and signaling in cells, which is mediated to a large extent by expression of US28."

**Category 2: Intervened Condition**

"Question": "The Bordetella adenylate cyclase toxin-hemolysin (CyaA) has multiple activities. In CD11b+ J774A.1 monocytes, does the CyaA-AC-toxoid, which is unable to generate cAMP, promote a faster, transient elevation of [Ca2+]i compared to intact CyaA?",

"Answer": "Yes"

"Question": "Dos/Gab family scaffolding adapters, including Gab1 and Gab2, are known to bind signal relay molecules and play a role in signal transduction. While mice lacking Gab1 die during embryogenesis, Gab2-/- mice are viable and generally healthy. However, the response of Gab2-/- mast cells to stimulation of the high affinity immunoglobulin-epsilon (IgE) receptor Fc(epsilon)RI

is defective. Are the responses of mast cells in Gab2-/- mice enhanced when stimulated by the high affinity immunoglobulin-epsilon (IgE) receptor Fc(epsilon)RI?",

"Answer": "No"

"Question": "Stromal cell-derived factor 1 alpha (SDF-1alpha) is a chemotactic factor for T lymphocytes and binds to the G-protein-coupled receptor CXCR4. What is the role of LIM kinase 1 (LIMK1) in the chemotaxis of T lymphocytes induced by SDF-1alpha?",

"Answer": "LIMK1 phosphorylates cofilin and regulates actin reorganization, playing a critical role in SDF-1alpha-induced chemotaxis of T lymphocytes."

"Question": "The CXC chemokine stromal cell-derived factor-1alpha (SDF-1) binds to CXCR4, a seven-transmembrane G protein-coupled receptor that plays a critical role in many physiological processes, including cell migration and cell fate decisions. CXCR4 is also implicated in various pathological conditions, such as metastatic spread and human immunodeficiency virus infection. In the context of SDF-1-induced cell migration in CXCR4-expressing cells, what is the role of Galpha(13) in the activation of Rho by CXCR4?",

"Answer": "Galpha(13) mediates the activation of Rho by CXCR4."

**Dimension 3: Investigation Target**

**Category 1: Single Component as Target**

"Question": "Advanced glycation end product (AGE) activation of the signal-transducing receptor for AGE (RAGE) has been linked to a proinflammatory phenotypic change within cells. Will human serum albumin modified with N(epsilon)-(carboxymethyl)lysine (CML) inhibit nuclear factor (NF)-kappaB-driven reporter gene expression in human monocytic THP-1 cells?",

"Answer": "No"

"Question": "Although the molecular mechanisms of hepatitis C virus (HCV) pathogenesis are not fully understood, the NS5A nonstructural protein of HCV has been found to interact with the growth factor receptor-bound protein 2 (Grb2) adaptor protein. To investigate the effects of NS5A on cellular signaling pathways, HeLa cells were stably expressing NS5A and were tested for their response to exogenous epidermal growth factor. Will HeLa cells stably expressing NS5A be refractory to ERK1/2 phosphorylation induced by exogenous epidermal growth factor?",

"Answer": "Yes"

"Question": "Enteropathogenic Escherichia coli (EPEC) and other related pathogens can trigger an early apoptotic response in host cells through the secretion of various effectors, including those from the type III secretion system. However, EPEC-infected cells do not typically progress to late apoptotic stages. What is the effect of NleH effectors, which are homologs of the Shigella effector kinase OspG, on caspase-3 activation during EPEC infection?",

"Answer": "NleH effectors inhibit caspase-3 activation during EPEC infection."

"Question": "Epithelial-mesenchymal transition (EMT) of tubular epithelial cells is a key event in renal interstitial fibrosis and the progression of chronic kidney disease (CKD). Apelin is a regulatory peptide involved in the regulation of normal renal hemodynamics and tubular functions. To examine the inhibitory effects of apelin on transforming growth factor-beta1 (TGF-beta1)-induced EMT in HK-2 cells, cells were co-treated with apelin and TGF-beta1. What is the effect of apelin on TGF-beta1-mediated upregulation of alpha-smooth muscle actin (alpha-SMA) and downregulation of E-cadherin in HK-2 cells?",

"Answer": "Apelin inhibits TGF-beta1-mediated upregulation of alpha-smooth muscle actin (alpha-SMA) and downregulation of E-cadherin in HK-2 cells."

**Category 2: Components Interaction as Target**

"Question": "Nucleotide-binding leucine-rich repeat-containing proteins, or NOD-like receptors (NLRs), are intracellular innate immune sensors that can regulate several signaling pathways, including MyD88- and TRIF-dependent pathways. NLRP12 is a member of the NLR family that can assemble into multimeric protein complexes known as inflammasomes. During infection with

Salmonella enterica serovar Typhimurium, does NLRP12 act as a negative regulator of the NFkappaB and MAPK signaling pathways?",

"Answer": "Yes"

"Question": "Pathogenic bacteria of the genus Yersinia employ a type III secretion system to inject bacterial effector proteins directly into the host cytosol. One of these effectors, the Yersinia serine/threonine protein kinase YpkA, is an essential virulence determinant involved in host actin cytoskeletal rearrangements and in inhibition of phagocytosis. Will Y. pseudotuberculosis expressing wild-type YpkA enhance Galphaq-mediated signaling pathways?",

"Answer": "No"

"Question": "Kaposi's sarcoma-associated herpesvirus (KSHV) has an immune evasion gene, K5. What is the effect of K5-mediated ubiquitylation on NKG2D ligands MHC class I-related chain A (MICA) and NK cell-mediated cytotoxicity?",

"Answer": "K5-mediated ubiquitylation signals internalization but not degradation of MICA and causes a potent reduction in NK cell-mediated cytotoxicity."

"Question": "Measles virus infection is characterized by virus-induced immune suppression that creates susceptibility to opportunistic infections, and it has been found that measles virus can inhibit cytokine responses by direct interference with host STAT protein-dependent signaling systems. The measles V protein plays a role in this interference, but how does the measles V protein specifically affect STAT protein-dependent signaling systems?",

"Answer": "The measles V protein interferes with STAT protein-dependent signaling systems by causing a defect in IFN-induced STAT nuclear accumulation."

**Category 3: Function as Target**

"Question": "The dysfunction of mitochondria has long been recognized as a key component in the progression of Parkinson's disease (PD). Can dysfunctional mitochondria lead to dysregulation of calcium homeostasis and raised mean intracellular calcium concentration in dopaminergic neurons?",

"Answer": "Yes"

"Question": "The gut epithelium self-renews every several days, providing an important innate defense system that limits bacterial colonization. However, Shigella efficiently colonizes the intestinal epithelium. Is the cell-cycle arrest caused by Shigella infection in HeLa cells independent of IpaB and Mad2L2?",

"Answer": "No"

"Question": "Enterohemorrhagic Escherichia coli (EHEC) is a diarrheagenic pathogen that employs a type III secretion system (T3SS) to translocate 50 effector proteins, which allow bacterial colonization and subversion of immune responses and disease progression. One of these effector proteins is EspW, which is found in various EHEC strains. What is the effect of deleting espW on cell morphology during EHEC infection?",

"Answer": "Infection of Swiss cells with an EHEC espW deletion mutant induces a cell shrinkage phenotype."

"Question": "Enteropathogenic and enterohaemorrhagic Escherichia coli (EPEC and EHEC) are food-borne pathogens that cause severe diarrhoeal disease in humans. Citrobacter rodentium is a related mouse pathogen that serves as a small animal model for EPEC and EHEC infections. What is the effect of an Non-LEE-encoded effector A (NleA) protein variant with greatly diminished interaction with all Sec24 paralogues on virulence in the mouse infection model?",

"Answer": "An NleA protein variant with greatly diminished interaction with all Sec24 paralogues does not confer virulence in the mouse infection model."

## A.2 FAILURE REASON CATEGORY CASES

We classify the error reason of biological pathway reasoning into four types: (1) **Unresolved Conclusion** For cases where the model fails to provide a definitive answer, indicating uncertainty or belief that the answer is unknown. (2) **Incomplete Answer** When the response lacks essential details, such as missing the requested effects or other key elements. (3) **Omission in Reasoning** For errors where critical pathway steps in the question's biological process are left out, causing the final answer to be incorrect. (4) **Faulty in Reasoning** When the reasoning path is correct, but there are significant errors in deducing the events within that pathway.

In this subsection, we provide examples of each failure reason type. The 'Question' and 'Answer' represent the input question and its labeled correct answer. The 'Wrong Answer by Model' refers to the incorrect response generated by the model. Additionally, we include a 'Reference Reasoning Process', which outlines the reasoning process leading to the correct final answer by LLaMA3.1 405B. This helps readers understand how errors occur during the reasoning process.

For the error types 'Omission in Reasoning' and 'Faulty Reasoning,' we present cases for both CoT and PathSeeker individually. These examples demonstrate how LLM reasoning fails either on its own or when augmented with a pathway graph.

**Category 1: Unresolved Conclusion**

---

"Question": "A large fraction of pediatric pre-B acute lymphoblastoid leukemias (ALL) consistently contain a t(1;19) chromosomal translocation, resulting in the production of a chimeric transcription factor containing the N-terminal transactivation domain of E2A fused to the C-terminal DNA-binding homeodomain of Pbx1. What is the effect of the E2A-Pbx1 fusion protein on the expression of the novel WNT gene, WNT-16, in pre-B ALL cells?",
"Answer": "The E2A-Pbx1 fusion protein activates the expression of WNT-16 in pre-B ALL cells.",

- - - - - - - - - - - - - - - - - - - - - - - - - - - - - - - - - - - - - - - - - - - -

"Wrong Answer by Model": "The E2A-Pbx1 fusion protein in pediatric pre-B acute lymphoblastic leukemias (ALL) with the t(1;19) chromosomal translocation is known to have oncogenic properties and can affect gene expression. In the case of the novel WNT gene, WNT-16, the E2A-Pbx1 fusion protein may have an effect on its expression in pre-B ALL cells. However, the specific effect of the fusion protein on WNT-16 expression would depend on the regulatory elements present in the WNT-16 gene and the transcriptional activity of the fusion protein. Further investigation would be needed to determine the exact effect of the E2A-Pbx1 fusion protein on WNT-16 expression in pre-B ALL cells.",

- - - - - - - - - - - - - - - - - - - - - - - - - - - - - - - - - - - - - - - - - - - -

"Reference Reasoning Process": "To determine the effect of the E2A-Pbx1 fusion protein on the expression of the novel WNT gene, WNT-16, in pre-B ALL cells, we can analyze the provided pathways for transcriptional regulation. The pathways indicate that both E2A (6929) and Pbx1 are transcription factors and that their fusion can lead to transcriptional activation of target genes.
Pathways 0 and 13 demonstrate that the E2A-Pbx1 fusion protein can cause the expression of WNT-16. The fusion of E2A and Pbx1 results in a chimeric transcription factor that retains the transactivation domain of E2A and the DNA-binding homeodomain of Pbx1.
Considering this information, it can be inferred that the E2A-Pbx1 fusion protein leads to increased expression of the novel WNT gene, WNT-16, in pre-B ALL cells. The fusion of E2A and Pbx1 results in a transcription factor that activates the expression of WNT-16, potentially contributing to the development or progression of pre-B ALL.

---

Figure 7: The Unresolved Conclusion error occurs when the model cannot provide a definitive answer, indicating uncertainty or the belief that the answer is unknown. In this case, the model is unable to determine the specific effect of the fusion protein on WNT-16 expression.

**Category 2: Incomplete Answer**

"Question": "Continual efferocytic clearance of apoptotic cells (ACs) by macrophages is crucial for injury resolution. The process of continual efferocytosis is enhanced by linking the metabolism of engulfed cargo from initial efferocytic events to subsequent rounds. How is continual efferocytosis enhanced in macrophages?",

"Answer": "Continual efferocytosis is enhanced in macrophages by the metabolism of apoptotic cell-derived arginine and ornithine to putrescine by macrophage arginase 1 (Arg1) and ornithine decarboxylase (ODC).",

- - - - - - - - - - - - - - - - - - - - - - - - - - - - - - - - - - - - - - - - - - - - - - - - -

"Wrong Answer by Model": "Continual efferocytosis is enhanced in macrophages through the process of metabolic reprogramming. When macrophages engulf apoptotic cells (ACs), they not only clear the ACs but also metabolize the engulfed cargo. This metabolic reprogramming involves the activation of specific metabolic pathways, such as glycolysis and fatty acid oxidation, to generate energy and metabolites necessary for the efficient clearance of ACs. By linking the metabolism of engulfed cargo from initial efferocytic events to subsequent rounds, macrophages can sustain their energy requirements and maintain the high phagocytic capacity needed for continual efferocytosis. This enhanced metabolic activity allows macrophages to efficiently clear ACs and promote injury resolution.",

- - - - - - - - - - - - - - - - - - - - - - - - - - - - - - - - - - - - - - - - - - - - - - - - -

"Reference Reasoning Process": "To determine how continual efferocytosis is enhanced in macrophages, we can analyze the provided pathways for the key steps involved in efferocytic clearance of apoptotic cells (ACs) and how they link to subsequent rounds.
The process starts with the engulfment of apoptotic cells. This involves the recognition of phosphatidylserine on the surface of apoptotic cells by receptors such as AGER (Pathway 0). This recognition leads to the activation of signaling pathways, including the activation of Rac family small GTPase 1 (RAC1), which is crucial for the engulfment process (Pathway 1).
Following engulfment, the metabolism of engulfed cargo plays a critical role in enhancing continual efferocytosis. Specifically, the pathway involving the solute carrier family 66 member 1 (SLC66A1) and the metabolism of L-Arginine (Pathway 6) leads to the production of ornithine. Ornithine then binds to ARG1 (Pathway 7), leading to the production of urea and more ornithine (Pathway 8). This process enhances continual efferocytosis by providing the necessary components for subsequent rounds of efferocytic clearance.
Moreover, the pathway involving the metabolism of ornithine to putrescine (Pathway 9) and then the activation of RAC1 by putrescine (Pathway 11) further supports the enhancement of continual efferocytosis. This is because RAC1 activation is crucial for the engulfment process, and its activation by putrescine ensures that the macrophage remains capable of engulfing subsequent apoptotic cells.
Therefore, based on the pathways provided, continual efferocytosis in macrophages is enhanced by the metabolism of engulfed cargo, specifically through the arginine and ornithine pathways, which ultimately lead to the activation of key signaling molecules such as RAC1 that are necessary for the efferocytic clearance of apoptotic cells.",

Figure 8: The Incomplete Answer error occurs when the response lacks essential details, such as missing the requested effects or other key elements. In this case, the model's response suggests that continual efferocytosis is enhanced in macrophages through metabolic reprogramming but fails to mention the metabolism of apoptotic cell-derived arginine and ornithine to putrescine, which is a key part of the labeled answer.

**Category 3: Omission in Reasoning**

**Omission in Reasoning of CoT**

"Question": "BAMBI (BMP and activin membrane-bound inhibitor) is known to negatively regulate the signaling activity of transforming growth factor-beta, activin, and BMP, and has been identified as a target of beta-catenin in colorectal and hepatocellular tumor cells. The canonical Wnt/beta-catenin pathway plays a crucial role in regulating embryogenesis

and tumorigenesis by promoting cell proliferation. Does BAMBI enhance or inhibit Wnt-promoted cell cycle progression?",
"Answer": "BAMBI increases Wnt-promoted cell cycle progression.",

- - - - - - - - - - - - - - - - - - - - - - - - - - - - - - - - - - - - - - - - - -

"Wrong Answer by Model": "BAMBI inhibits Wnt-promoted cell cycle progression. The canonical Wnt/beta-catenin pathway promotes cell proliferation, and BAMBI has been identified as a target of beta-catenin. BAMBI negatively regulates the signaling activity of transforming growth factor-beta, activin, and BMP, which are involved in cell cycle regulation. Therefore, BAMBI acts as a negative regulator of the Wnt pathway and inhibits Wnt-promoted cell cycle progression.",

- - - - - - - - - - - - - - - - - - - - - - - - - - - - - - - - - - - - - - - - - -

"Reference Reasoning Process": "To determine whether BAMBI enhances or inhibits Wnt-promoted cell cycle progression, we need to analyze its interactions with key components of the Wnt signaling pathway.
From pathways 10, 11, and 12, we can see that BAMBI (BMP and activin membrane-bound inhibitor) interacts with LRP6, FZD10, and DVL1, which are all crucial components of the Wnt signaling pathway. These interactions are described as "activation" and "binding/association," which suggests that BAMBI may be enhancing the activity of these components.
However, it's also known that BAMBI is a negative regulator of the TGF-beta and BMP signaling pathways, which can intersect with the Wnt signaling pathway.
Considering the complexity of these interactions and the multiple pathways involved, we need to look for more direct evidence of BAMBI's effect on Wnt-promoted cell cycle progression.
Given the information provided, it seems that BAMBI's interactions with the Wnt signaling pathway are more likely to be enhancing or modulating rather than inhibiting. However, without more direct evidence, it's difficult to make a definitive conclusion.
Therefore, based on the available information, it appears that BAMBI may enhance Wnt-promoted cell cycle progression, but further research is needed to confirm this conclusion.",

Figure 9: Omission in Reasoning refers to errors where critical steps in the biological pathway described in the question are omitted, leading to an incorrect final answer. In this case, the model's reasoning process only considered that BAMBI has been identified as a target of beta-catenin but overlooked the pathway in which BAMBI interacts with LRP6, FZD10, and DVL1, all of which are crucial components of the Wnt signaling pathway. As a result, the reasoning led to an incorrect conclusion.

**Omission in Reasoning of PathSeeker**

"Question": "Angiotensin II (Ang II) is an octapeptide that plays a crucial role in the cardiovascular system, and its formation and degradation are tightly regulated. To investigate the role of mast cell proteases in Ang II production, ex vivo cultures of peritoneal cells from mouse strains lacking mast cell heparin or mouse mast cell protease 4 (mMCP-4) were used. What is the effect of heparin deficiency on the formation and degradation of Ang II in these peritoneal cell cultures?",
"Answer": "The formation and degradation of Ang II are strongly reduced in heparin-deficient peritoneal cells.",

- - - - - - - - - - - - - - - - - - - - - - - - - - - - - - - - - - - - - - - - - -

"Wrong Answer by Model": "Based on the provided pathways, the effect of heparin deficiency on the formation and degradation of Ang II in peritoneal cell cultures can be inferred. Heparin is known to play a role in the regulation of Ang II production.
Pathway 0 indicates that carboxypeptidase A3 (CPA3) is involved in the degradation of Ang II. This suggests that heparin deficiency may lead to a decrease in the degradation of Ang II, potentially resulting in increased levels of Ang II.

Pathway 1 shows that angiotensin I converting enzyme (ACE) is responsible for the formation of Ang II. However, this pathway does not directly provide information about the effect of heparin deficiency on Ang II formation.

Considering the information from pathways 0 and 1, it can be inferred that heparin deficiency may primarily affect the degradation of Ang II rather than its formation. Without heparin, the degradation of Ang II by carboxypeptidase A3 may be impaired, leading to increased levels of Ang II in the peritoneal cell cultures.",

- - - - - - - - - - - - - - - - - - - - - - - - - - - - - - - - - - - - - - - - - - - - - - -

"Reference Reasoning Process": "To determine the effect of heparin deficiency on the formation and degradation of Ang II in peritoneal cell cultures, we can analyze the provided pathways for the role of mast cell proteases in Ang II production.

The pathways provided do not directly mention heparin or its deficiency. However, we can infer the potential effects of heparin deficiency by understanding the role of mast cell proteases in Ang II production.

Pathway 1 shows the conversion of Angiotensin I to Angiotensin III. Pathway 2 demonstrates the binding of Angiotensin I to angiotensin I converting enzyme (ACE), which is involved in the conversion of Angiotensin I to Angiotensin II. Pathway 3 shows the activation of angiotensin II receptor type 2 by Angiotensin II. Pathway 4 demonstrates the activation of AGTR1 and AGTR2 receptors by Angiotensin II. Pathway 5 shows the activation of angiotensin II receptor type 2 by Angiotensin A.

Considering the information provided by these pathways, we can infer that heparin deficiency may affect the activity of mast cell proteases, such as mouse mast cell protease 4 (mMCP-4), which are involved in the conversion of Angiotensin I to Angiotensin II. Heparin is a cofactor for the activation of mast cell proteases, and its deficiency may lead to reduced activity of these proteases.

Therefore, the effect of heparin deficiency on the formation and degradation of Ang II in peritoneal cell cultures may be a reduction in the conversion of Angiotensin I to Angiotensin II, leading to decreased levels of Angiotensin II.",

Figure 10: Omission in Reasoning refers to errors where critical steps in the biological pathway involved in the question are omitted, leading to an incorrect final answer. In this case, the model's reasoning process considered the pathway involving the degradation of Ang II but missed the more important pathway of the conversion of Angiotensin I to Angiotensin II. This resulted in reasoning that led to an incorrect conclusion.

**Category 4: Faulty in Reasoning**

**Faulty in Reasoning of CoT**

"Question": "Enteropathogenic Escherichia coli (EPEC) is a major causative agent of infant diarrhoea in developing countries, and the EspF effector protein is injected from EPEC into host cells via a type III secretion system, disrupting host intestinal barrier function and initiating the mitochondrial death pathway. To clarify the manner in which EspF affects host cells, eukaryotic EspF-binding proteins were identified using affinity purification, and Abcf2, a protein of unknown function and member of the ABC-transporter family, was found to bind EspF. What is the effect of knocking down Abcf2 expression on EspF-induced caspase cleavage?",

"Answer": "Knocking down Abcf2 expression increases EspF-induced caspase 9 and caspase 3 cleavage.",

- - - - - - - - - - - - - - - - - - - - - - - - - - - - - - - - - - - - - - - - - - - - - - - - - - - -

"Wrong Answer by Model": "The effect of knocking down Abcf2 expression on EspF-induced caspase cleavage is likely to decrease or inhibit caspase cleavage. Abcf2, as a binding protein of EspF, may play a role in the signaling pathway that leads to caspase activation and subsequent cleavage. By knocking down Abcf2 expression, the interaction between EspF and Abcf2 is disrupted, potentially preventing or reducing the activation of caspases and subsequent cleavage. This suggests that Abcf2 is involved in the EspF-induced mitochondrial death pathway and its knockdown may have a protective effect against the disruption of host intestinal barrier function caused by EPEC infection.",

- - - - - - - - - - - - - - - - - - - - - - - - - - - - - - - - - - - - - - - - - - - - - - - - - - - -

"Reference Reasoning Process": "To determine the effect of knocking down Abcf2 expression on EspF-induced caspase cleavage, we should analyze the provided pathways for the interactions between EspF, Abcf2, and caspase.
Pathway 0 provides information that EspF binds Abcf2, and this interaction inhibits Abcf2. Additionally, pathway 2 reveals that Abcf2 inhibits caspase 9, which is involved in the intrinsic apoptotic pathway.
When EspF binds Abcf2, it inhibits Abcf2's function. Since Abcf2 normally inhibits caspase 9, the inhibition of Abcf2 by EspF would lead to an increase in caspase 9 activity.
If Abcf2 expression is knocked down, its inhibitory effect on caspase 9 would be reduced. As a result, caspase 9 activity would increase, similar to the effect of EspF binding Abcf2.
Therefore, knocking down Abcf2 expression would enhance EspF-induced caspase cleavage. The reduction in Abcf2's inhibitory effect on caspase 9 would amplify the pro-apoptotic signal, leading to increased caspase cleavage.
In summary, the effect of knocking down Abcf2 expression on EspF-induced caspase cleavage would be an increase in caspase cleavage due to the reduced inhibitory effect of Abcf2 on caspase 9.",

Figure 11: Faulty Reasoning occurs when the reasoning path is correct, but there are significant errors in deducing the events within that pathway. In this case, the model's reasoning step proposes that EspF binds to Abcf2 and activates it, which is incorrect, as it actually inhibits Abcf2. As a result, the reasoning leads to an incorrect conclusion.

**Faulty in Reasoning of PathSeeker**

"Question": "Bacterial pathogens often manipulate host immune pathways to establish acute and chronic infection, and many Gram-negative bacteria do this by secreting effector proteins through a type III secretion system that alter the host response to the pathogen. In the case of Salmonella, the phage-encoded GogB effector protein targets the host SCF E3 type ubiquitin ligase through an interaction with Skp1 and the human F-box only 22 (FBXO22) protein. What is the effect of GogB-deficient Salmonella on tissue damage and colonization in the gut during long-term chronic infections?",

"Answer": "GogB-deficient Salmonella cause extensive tissue damage and enhanced colonization in the gut during long-term chronic infections.",

- - - - - - - - - - - - - - - - - - - - - - - - - - - - - - - - - - - - - - - - - - - - - - - - - - - -

"Wrong Answer by Model": "Based on the provided pathways, the effect of GogB-deficient Salmonella on tissue damage and colonization in the gut during long-term chronic infections can be inferred.

Pathway 0 indicates that the GogB effector protein targets the host SCF E3 type ubiquitin ligase through an interaction with Skp1 and the human F-box only 22 (FBXO22) protein. This pathway is associated with Salmonella infection.

Pathways 1 and 2 provide information about other type III secretion system effectors (SseK1 and SseK3) and their interactions with TNF-NFKB signaling pathways. These pathways are also linked to Salmonella infection.

Pathway 3 shows the involvement of tumor necrosis factor (TNF) and its receptor TNFRSF1A in various signaling pathways, including Salmonella infection.

Pathway 4 indicates the activation of NFKB, a transcription factor involved in inflammation and immune responses, by TNF. This pathway is associated with Salmonella infection.

Pathways 5 and 6 provide information about the activation and inhibition of NFKB inhibitor alpha (NFKBIA) and its interaction with NFKB. These pathways are also linked to Salmonella infection.

Pathway 7 shows the activation of MAPK8 (JNK) and its phosphorylation by FOS, which is associated with Salmonella infection.

Based on the provided pathways, it can be inferred that GogB-deficient Salmonella would have a reduced ability to manipulate host immune pathways and alter the host response to the pathogen. This could result in decreased tissue damage and colonization in the gut during long-term chronic infections. However, the specific effects on tissue damage and colonization would require further investigation.",
------------------------------------------------------------
"Reference Reasoning Process": "Based on the pathways provided, the effect of GogB-deficient Salmonella on tissue damage and colonization in the gut during long-term chronic infections can be inferred.

GogB is a phage-encoded effector protein that targets the host SCF E3 type ubiquitin ligase through an interaction with Skp1 and the human F-box only 22 (FBXO22) protein (Pathway 0). This interaction inhibits the NF-kappaB signaling pathway, which is a crucial pathway involved in inflammation and immune response.

In the absence of GogB, the inhibition of the NF-kappaB signaling pathway is lifted, leading to an increase in the production of pro-inflammatory cytokines, such as TNF-alpha (Pathway 3). This increase in inflammation can lead to tissue damage and exacerbate the symptoms of the infection.

Furthermore, the NF-kappaB signaling pathway is also involved in the regulation of apoptosis (programmed cell death). The inhibition of this pathway by GogB can lead to a decrease in apoptosis, allowing the infected cells to survive and potentially leading to persistent infection (Pathway 6).

Additionally, GogB-deficient Salmonella may also have altered interactions with other host proteins, such as SifA, PipB2, and SopD2, which are involved in regulating microtubule dynamics and transport (Pathways 7, 9, and 10). These altered interactions can affect the ability of Salmonella to colonize and persist in the gut.

Overall, the loss of GogB in Salmonella is likely to lead to increased tissue damage and inflammation during long-term chronic infections, as well as altered interactions with host proteins that can affect the ability of the bacteria to colonize and persist in the gut.

In conclusion, the effect of GogB-deficient Salmonella on tissue damage and colonization in the gut during long-term chronic infections is likely to be an increase in tissue damage and inflammation, as well as altered interactions with host proteins that can affect the ability of the bacteria to colonize and persist in the gut.",

Figure 12: Faulty Reasoning occurs when the reasoning path is correct, but there are significant errors in deducing the events within that pathway. In this case, the model reasoned along the NF-kappaB signaling pathway but failed to deduce that a GogB deficiency leads to an increase in inflammation. Consequently, the final conclusion is incorrect.

## A.3 DATA CREATION AND FILTER PIPELINE

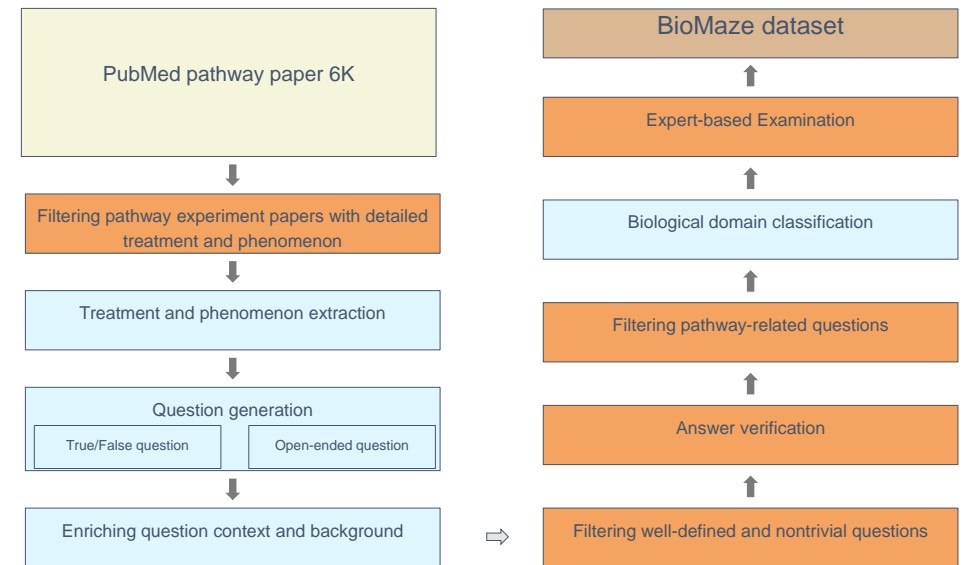

Figure 13: Dataset BioMaze creation pipeline.

The overall dataset creation pipeline is shown in Figure 13.

To ensure question quality, we employ a two-step process. First, we create and filter questions using an advanced language model ( LLaMa 3.1-405B) to assess their relevance and clarity. Subsequently, each question undergoes a final quality check by human experts.

The well-define filter removes questions that are poorly defined, unpredictable (e.g., asking for specific measurement values), or require more than one prediction, and the nontrivial filter removes the data that answers revealed in the question's context.

To validate the answer quality, we require the LLM (LLaMa 3.1-405B) to answer the questions based on the original paper's content. The model is explicitly instructed to respond with Undetermined if it cannot confidently generate an answer. Each question is tested five times, and only questions that are consistently answered correctly (i.e., aligned with the intended label) and not marked as Undetermined in any of the trials are retained. This process helps eliminate questions with incorrect labels, ambiguous phrasing, or poor structure.

In the final stage, human experts perform an additional quality check to refine the questions further. Approximately 5% of the data is filtered out at this stage, primarily due to issues such as hint leakage in the question, overly complex phrasing (e.g., asking for multiple facts), or poorly defined structure. During this stage, human reviewers also verify label correctness, ensuring the dataset's overall reliability and usability.

## A.4 QUESTION KEY WORDS DISTRIBUTION

We present the distribution of question keywords in Figure 14. While these keywords do not directly correspond to the three main categories we primarily use, they offer an additional perspective on the dataset. Below are explanations of the keywords:

General Influence Inquiry: Can x influence y or not?

Activation Inquiry: Can x activate y?

Inhibition Inquiry: Can x inhibit y?

Dependency Inquiry: Is y dependent on x?

Induction Question: Can x induce y?

Relief Inquiry: Can x relieve y?

Mechanism Question: Does x influence y via a specific mechanism?

Exclusive Mechanism Question: Is a specific mechanism the only mechanism for process z?

Significance Inquiry: Will x cause a significant/insignificant phenomenon y?

Baseline Comparison Question: Is x different from the baseline?

Experimental Observation Question: Will a specific detailed phenomenon be observed in the experiment?

Physiological Observation Question: Will a specific phenomenon be observed in the body?

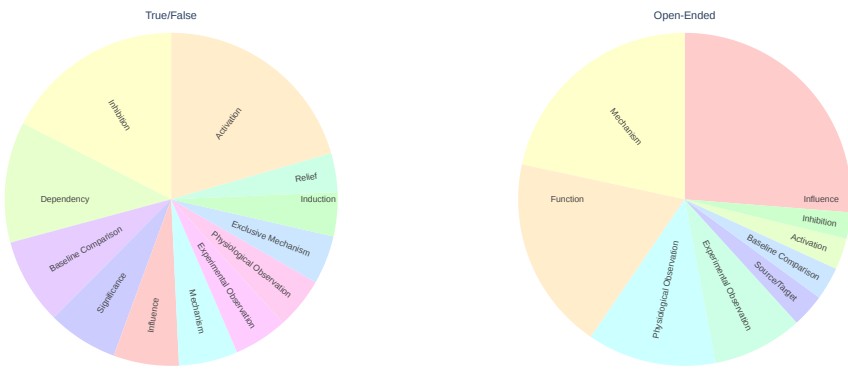

Figure 14: Dataset key words distribution.

### A.5 PATHWAY GRAPH DATABASE STATISTICS

Table 7: Data statistic of our pathway network database.

| Entries | Edges | Involved biological process |
|---------|-------|------------------------------|
| 8939    | 15131 | 2265                         |

### A.6 IMPLEMENTATION OF SUBGRAPH RETRIEVER

We define the subgraph retriever as follows:

$$\text{Search\_Subgraph}(query, N) = \underset{S \subseteq P, S \text{ is connected}, |S|=N}{\text{argmax}} \sum_{i \in V_S \cup E_S} \text{score}(i, query) \tag{7}$$

which is hard to solve directly for huge graph databse. Inspire by He et al. (2024) that convert optimal subgraph problem into a Prize-Collecting Steiner Tree (PCST) problem, we solve the problem as a bi-level optimization as follows:

$$S(C_e) = \underset{S \subseteq P, S \text{ is connected}}{\text{argmax}} \sum_{i \in V_S \cup E_S} \text{score}(i, query) - |E_S| \times C_e$$
$$C_e^* = \underset{C_e}{\text{argmin}} ||S(C_e)| - N| \tag{8}$$

Then

$$\text{Search\_Subgraph}(query, N) = S(C_e^*)$$

The inner optimization problem, a Prize-Collecting Steiner Tree (PCST) problem, seeks to identify a connected subgraph that maximizes the total score of its nodes and edges while minimizing the edge costs, as utilized in (He et al., 2024). The outer optimization problem minimizes the difference between the optimal subgraph size and the target size. In practice, we apply a fast-solving method for the inner problem proposed by (Hegde et al., 2015) and employ a binary root search algorithm for the outer problem.

## A.7 PATHSEEKER CASE

Please explore pathways to find relevant information regarding the following question: Worldwide, one of the most prevalent forms of chronic disease is alcoholic fatty liver, which may progress to more severe forms of liver injury. Chronic ethanol exposure is thought to cause enhanced hepatic lipogenesis and impaired fatty acid oxidation. Adiponectin is an adipose-derived hormone that has been associated with the development of alcoholic liver steatosis in rodent models. Does adiponectin have a protective role against alcoholic liver steatosis?

Action: Thought: To find information regarding the role of adiponectin in protecting against alcoholic liver steatosis, I can use the search_biopathway_subgraph_global tool to explore relevant biological pathways. I should convert all the question content into keywords to enhance retrieval quality. Action:
    search_biopathway_subgraph_global(['Adiponectin,
    adipose-derived hormone, associated with the development of
    alcoholic liver steatosis in rodent models, protective role
    against alcoholic liver steatosis, chronic ethanol exposure,
    enhanced hepatic lipogenesis, impaired fatty acid oxidation,
    liver injury'])
    End Action

State: 0) C00469: Ethanol; Ethyl alcohol; Methylcarbinol |
406999: microRNA 217 MIR217 MIRN217 mir-217 | PCrel indirect
effect activation | hsa04936: Alcoholic liver disease
1) 406999: microRNA 217 MIR217 MIRN217 mir-217 | 23411: sirtuin 1
SIRT1 SIR2 SIR2L1 SIR2alpha | PPrel inhibition indirect effect |
hsa04936: Alcoholic liver disease
2) 406999: microRNA 217 MIR217 MIRN217 mir-217 | 29896 6434:
TRA2A AWMS1 HSU53209 | PPrel inhibition indirect effect |
hsa04936: Alcoholic liver disease
3) 29896 6434: TRA2A AWMS1 HSU53209 | 23175 64900 9663: LPIN1
PAP1 | PPrel activation | hsa04936: Alcoholic liver disease
4) 23175 64900 9663: LPIN1 PAP1 | 6720: sterol regulatory element
binding transcription factor 1 SREBF1 HMD IFAP2 SREBP1 bHLHd1 |
PPrel inhibition | hsa04936: Alcoholic liver disease
5) 6720: sterol regulatory element binding transcription factor 1
SREBF1 HMD IFAP2 SREBP1 bHLHd1 | 6319 79966: SCD FADS5 MSTP008
SCD1 SCDOS hSCD1 | GErel expression | hsa04152: AMPK signaling
pathway hsa04936: Alcoholic liver disease
6) 6720: sterol regulatory element binding transcription factor 1
SREBF1 HMD IFAP2 SREBP1 bHLHd1 | 23175 64900 9663: LPIN1 PAP1 |
GErel expression | hsa04936: Alcoholic liver disease
7) C00469: Ethanol; Ethyl alcohol; Methylcarbinol | 51094 79602:
ADIPOR1 ACDCR1 CGI-45 CGI45 PAQR1 TESBP1A | PCrel inhibition
indirect effect | hsa04936: Alcoholic liver disease
8) C00469: Ethanol; Ethyl alcohol; Methylcarbinol | 51422 53632
5562 5563 5564 5565 5571: PRKAG2 AAKG AAKG2 CMH6 H91620p WPWS |

PCrel inhibition indirect effect | hsa04936: Alcoholic liver disease
9) C00469: Ethanol; Ethyl alcohol; Methylcarbinol | 50507: NADPH oxidase 4 NOX4 KOX KOX-1 RENOX | PCrel activation | hsa04936: Alcoholic liver disease
10) 50507: NADPH oxidase 4 NOX4 KOX KOX-1 RENOX | C22381: Reactive oxygen species; ROS | PCrel activation | hsa04936: Alcoholic liver disease
11) C22381: Reactive oxygen species; ROS | 51422 53632 5562 5563 5564 5565 5571: PRKAG2 AAKG AAKG2 CMH6 H91620p WPWS | PCrel inhibition indirect effect | hsa04936: Alcoholic liver disease
12) C22381: Reactive oxygen species; ROS | 10000 207 208: AKT3 MPPH MPPH2 PKB-GAMMA PKBG PRKBG RAC-PK-gamma RAC-gamma STK-2 | PCrel inhibition | hsa04936: Alcoholic liver disease
13) 10000 207 208: AKT3 MPPH MPPH2 PKB-GAMMA PKBG PRKBG RAC-PK-gamma RAC-gamma STK-2 | 1499: catenin beta 1 CTNNB1 CTNNB EVR7 MRD19 NEDSDV armadillo | PPrel activation phosphorylation | hsa04936: Alcoholic liver disease
14) C22381: Reactive oxygen species; ROS | 5606 5608: MAP2K3 MAPKK3 MEK3 MKK3 PRKMK3 SAPKK-2 SAPKK2 | PCrel activation | hsa04936: Alcoholic liver disease
15) C00469: Ethanol; Ethyl alcohol; Methylcarbinol | 712 713 714: C1QA C1QD1 | PCrel activation | hsa04936: Alcoholic liver disease
16) C00469: Ethanol; Ethyl alcohol; Methylcarbinol | 3605: interleukin 17A IL17A CTLA-8 CTLA8 IL-17 IL-17A IL17 ILA17 | PCrel indirect effect activation | hsa04936: Alcoholic liver disease
17) C00084: Acetaldehyde; Ethanal | 51422 53632 5562 5563 5564 5565 5571: PRKAG2 AAKG AAKG2 CMH6 H91620p WPWS | PCrel inhibition indirect effect | hsa04936: Alcoholic liver disease
18) 9370: adiponectin C1Q and collagen domain containing ADIPOQ ACDC ACRP30 ADIPQTL1 ADPN APM-1 APM1 GBP28 | 51094 79602: ADIPOR1 ACDCR1 CGI-45 CGI45 PAQR1 TESBP1A | PPrel activate activation | hsa04152: AMPK signaling pathway hsa04211: Longevity regulating pathway hsa04936: Alcoholic liver disease hsa04920: Adipocytokine signaling pathway N01788: ADIPOQ-ADIPOR signaling pathway
19) 7132: TNF receptor superfamily member 1A TNFRSF1A CD120a FPF TBP1 TNF-R TNF-R-I TNF-R55 TNFAR TNFR1 TNFR55 TNFR60 p55 p55-R p60 | 6720: sterol regulatory element binding transcription factor 1 SREBF1 HMD IFAP2 SREBP1 bHLHd1 | PPrel indirect effect activation | hsa04936: Alcoholic liver disease

Action: Thought: I have gathered a large number of pathways related to the question. To better understand the role of adiponectin in protecting against alcoholic liver steatosis, I will check the neighboring pathways of Pathway 18, which is directly related to adiponectin. I will use the function search_biopathway_triple_N_hop_subgraph with history_line_id set to 18. The keywords are the same as previously used.
    Action:
    search_biopathway_triple_N_hop_subgraph(18, ['Adiponectin, adipose-derived hormone, associated with the development of alcoholic liver steatosis in rodent models, protective role against alcoholic liver steatosis, chronic ethanol exposure, enhanced hepatic lipogenesis, impaired fatty acid oxidation, liver injury'])
    End Action

State: 20) C00219: Arachidonate; Arachidonic acid;
(5Z8Z11Z14Z)-Icosatetraenoic acid; cis-581114-Eicosatetraenoic
acid; 5Z8Z11Z14Z-Eicosatetraenoic acid; | 2180 2181 2182 23205
23305 51703: ACSL1 ACS1 FACL1 FACL2 LACS LACS1 LACS2 | pathway
bind to | hsa04216: Ferroptosis N01590: Arachidonate/Adrenic acid
metabolism
21) 2180 2181 2182 23205 23305 51703 81616: ACSL1 ACS1 FACL1
FACL2 LACS LACS1 LACS2 | C00162: Fatty acid | PCrel  | hsa04920:
Adipocytokine signaling pathway
22) 1376: carnitine palmitoyltransferase 2 CPT2 CPT1 CPTASE IIAE4
| 51 8310: ACOX1 ACOX AOX MITCH PALMCOX SCOX | ECrel compound |
hsa00071: Fatty acid degradation
23) 1376: carnitine palmitoyltransferase 2 CPT2 CPT1 CPTASE IIAE4
| 126129 1374 1375: CPT1C CATL1 CPT1-B CPT1P CPTI-B CPTIC SPG73 |
ECrel compound | hsa00071: Fatty acid degradation
24) 126129 1374 1375: CPT1C CATL1 CPT1-B CPT1P CPTI-B CPTIC SPG73
| 2180 2181 2182 23205 23305 51703 81616: ACSL1 ACS1 FACL1 FACL2
LACS LACS1 LACS2 | ECrel compound | hsa00071: Fatty acid
degradation hsa04920: Adipocytokine signaling pathway
25) 4217: mitogen-activated protein kinase kinase kinase 5 MAP3K5
ASK1 MAPKKK5 MEKK5 | 5609 6416: MAP2K7 JNKK2 MAPKK7 MEK MEK_7
MKK7 PRKMK7 SAPKK-4 SAPKK4 | PPrel activate activation
phosphorylation | hsa05418: Fluid shear stress and
atherosclerosis hsa04668: TNF signaling pathway hsa04936:
Alcoholic liver disease hsa05208: Chemical carcinogenesis -
reactive oxygen species N01407: Metals to JNK signaling pathway
26) 5609 6416: MAP2K7 JNKK2 MAPKK7 MEK MEK_7 MKK7 PRKMK7 SAPKK-4
SAPKK4 | 5599 5601 5602: MAPK8 JNK JNK-46 JNK1 JNK1A2 JNK21B1/2
PRKM8 | PPrel activate activation phosphorylation | hsa05418:
Fluid shear stress and atherosclerosis hsa05135: Yersinia
infection hsa05417: Lipid and atherosclerosis hsa05167: Kaposi
sarcoma-associated herpesvirus infection hsa04620: Toll-like
receptor signaling pathway hsa04668: TNF signaling pathway
27) 5599 5601 5602: MAPK8 JNK JNK-46 JNK1 JNK1A2 JNK21B1/2 PRKM8
SAPK1 SAPK1c | 3667 8471 8660: IRS1 HIRS-1 | PPrel inhibition
phosphorylation | hsa04910: Insulin signaling pathway hsa05010:
Alzheimer disease hsa04930: Type II diabetes mellitus hsa04920:
Adipocytokine signaling pathway
28) 5599 5601 5602: MAPK8 JNK JNK-46 JNK1 JNK1A2 JNK21B1/2 PRKM8
SAPK1 SAPK1c | C00162: Fatty acid | PCrel  | hsa04930: Type II
diabetes mellitus
29) 4790 5970: NFKB1 CVID12 EBP-1 KBF1 NF-kB NF-kB1 NF-kappa-B1
NF-kappaB NF-kappabeta NFKB-p105 | 2919 2920 2921: CXCL1 FSP GRO1
GROa MGSA MGSA-a NAP-3 SCYB1 | PPrel activation expression |
hsa04621: NOD-like receptor signaling pathway hsa05167: Kaposi
sarcoma-associated herpesvirus infection hsa05417: Lipid and
atherosclerosis hsa05120: Epithelial cell signaling in
Helicobacter pylori infection hsa04936: Alcoholic liver disease
30) 4790 5970: NFKB1 CVID12 EBP-1 KBF1 NF-kB NF-kB1 NF-kappa-B1
NF-kappaB NF-kappabeta NFKB-p105 NFKB-p50 | 4792: NFKB inhibitor
alpha NFKBIA EDAID2 IKBA MAD-3 NFKBI | PPrel missing interaction
dissociation | hsa05215: Prostate cancer hsa05167: Kaposi
sarcoma-associated herpesvirus infection hsa05161: Hepatitis B
hsa05220: Chronic myeloid leukemia hsa05160: Hepatitis C
hsa04936: Alcoholic liver disease hsa04657: IL-17 signaling
pathway
31) 4790 5970: NFKB1 CVID12 EBP-1 KBF1 NF-kB NF-kB1 NF-kappa-B1
NF-kappaB NF-kappabeta NFKB-p105 NFKB-p50 NFkappaB | 9021:
suppressor of cytokine signaling 3 SOCS3 ATOD4 CIS3 Cish3 SOCS-3

```
SSI-3 SSI3 | GErel expression | hsa04920: Adipocytokine signaling
pathway
32) 9021: suppressor of cytokine signaling 3 SOCS3 ATOD4 CIS3
Cish3 SOCS-3 SSI-3 SSI3 | 3667 8471 8660: IRS1 HIRS-1 | PPrel
inhibition | hsa04920: Adipocytokine signaling pathway
33) 51094 79602: ADIPOR1 ACDCR1 CGI-45 CGI45 PAQR1 TESBP1A |
C00076: Calcium cation; Ca2+; Calcium(2+); Calcium ion | PCrel
indirect effect activation | hsa04152: AMPK signaling pathway
hsa04936: Alcoholic liver disease
34) C00076: Calcium cation; Ca2+; Calcium(2+); Calcium ion |
10645: calcium/calmodulin dependent protein kinase kinase 2
CAMKK2 CAMKK CAMKKB | PCrel activation | hsa04152: AMPK signaling
pathway hsa04211: Longevity regulating pathway hsa04936:
Alcoholic liver disease hsa04920: Adipocytokine signaling pathway
35) 10645: calcium/calmodulin dependent protein kinase kinase 2
CAMKK2 CAMKK CAMKKB | 51422 53632 5562 5563 5564 5565 5571:
PRKAG2 AAKG AAKG2 CMH6 H91620p WPWS | PPrel activation
phosphorylation | hsa04152: AMPK signaling pathway hsa04921:
Oxytocin signaling pathway hsa04211: Longevity regulating pathway
hsa04936: Alcoholic liver disease hsa04920: Adipocytokine
signaling pathway
36) 51094 79602: ADIPOR1 ACDCR1 CGI-45 CGI45 PAQR1 TESBP1A |
5465: peroxisome proliferator activated receptor alpha PPARA
NR1C1 PPAR PPAR-alpha PPARalpha hPPAR | PPrel indirect effect
activation | hsa04920: Adipocytokine signaling pathway
37) 5465: peroxisome proliferator activated receptor alpha PPARA
NR1C1 PPAR PPAR-alpha PPARalpha hPPAR | 126129 1374 1375: CPT1C
CATL1 CPT1-B CPT1P CPTI-B CPTIC SPG73 | GErel expression |
hsa04920: Adipocytokine signaling pathway
38) 5465: peroxisome proliferator activated receptor alpha PPARA
NR1C1 PPAR PPAR-alpha PPARalpha hPPAR | C00638: Long-chain fatty
acid; Higher fatty acid | PCrel  | hsa04920: Adipocytokine
signaling pathway

Action: Thought: I have gathered a large number of pathways
related to the question. To better understand the role of
adiponectin in protecting against alcoholic liver steatosis, I
will check the neighboring pathways of Pathway 18, which is
directly related to adiponectin. I will use the function
search_biopathway_triple_N_hop_subgraph with history_line_id set
to 18. The keywords are the same as previously used.
    Action:
    search_biopathway_triple_N_hop_subgraph(18, ['Adiponectin,
    adipose-derived hormone, associated with the development of
    alcoholic liver steatosis in rodent models, protective role
    against alcoholic liver steatosis, chronic ethanol exposure,
    enhanced hepatic lipogenesis, impaired fatty acid oxidation,
    liver injury'])
    End Action

State: No new pathways were found besides those previously seen.

Action: Thought: Since no new pathways were found, I will finish
the exploration.
Answer:
Finished.

State: You finished the task.
```

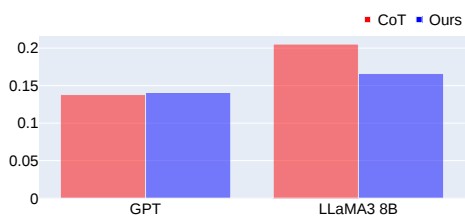

Figure 15: Uncertainty measured by entropy.

## A.8 BASELINE DETAILS

**Chain-of-Thought (CoT)** (Wei et al., 2022; Kojima et al., 2022) prompting encourages language models to generate step-by-step reasoning to solve tasks. We adopt CoT as the unaugmented standard method for LLM reasoning.

**Chain-of-Knowledge (CoK)** (Li et al., 2023a) enhances chain-of-thought reasoning through knowledge verification, enabling language models to handle knowledge-intensive tasks. CoK first generates reasoning using chain-of-thought processes, then employs knowledge triples to verify the accuracy of the reasoning. While the reasoning of CoK is primarily driven by the language model, graph-based information is used for fact verification.

**Think-of-Graph (ToG)** (Sun et al., 2023) is an interactive reasoning method designed to actively navigate knowledge graphs for question solving. It primarily uses large language models to prune knowledge graph edges, thereby enabling efficient knowledge acquisition from complex graphs. The reasoning process in ToG is guided by graph navigation.

**G-Retriever** (He et al., 2024) is a graph retriever-augmented generation method that retrieves relevant subgraphs from a database and generates answers based on the retrieved subgraphs. While the original model in their work uses a graph encoder to encode graph data as a separate modality, in this work, we directly implement the graph-to-text encoder for improved versatility and better comparability with other methods.

## A.9 UNCERTAINTY MEASURE

**Uncertainty** We investigate whether graph augmentation can reduce the uncertainty in model responses. We measure this uncertainty in the discriminant task by calculating five times the entropy of the final results, as shown in Figure 15. Notably, graph augmentation reduces prediction uncertainty for LLaMA3-8B but not for GPT-3.5. This discrepancy may arise because GPT-3.5 tends to have fewer hallucinations, whereas LLaMA3-8B may exhibit overconfidence in some generations where it is uncertain.

## A.10 EVALUATION OF THE EVALUATOR QUALITY

As the generation task involves LLMs as evaluators, we assess the quality of the evaluation method by comparing the result with the human manual annotation score. The accuracy of LLaMA3.1 405B with human annotation is 96%, while the inconsistency lines in the case that the answer is close to the ground truth but the expression is general and missing details.

## A.11 PERFORMANCE ON BIOLOGICAL DOMAINS

Figure 16 illustrates both GPT-3.5 and LLaMA3 8B performance on BioMaze's different biological domains.

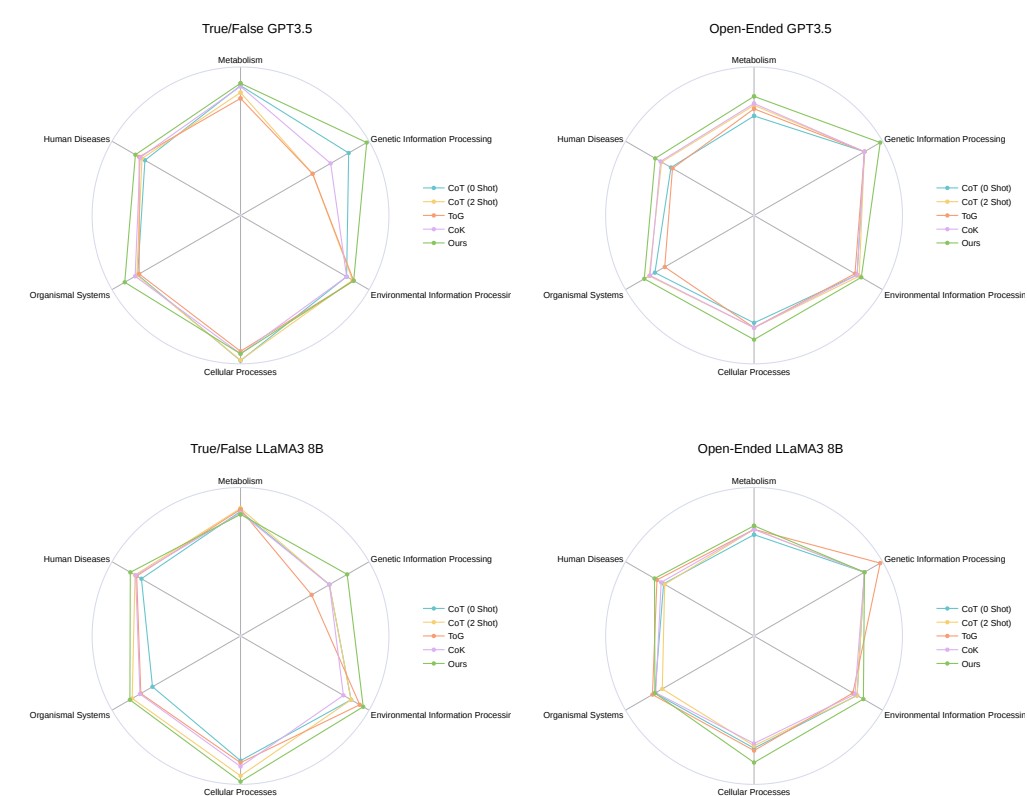

Figure 16: The radar chart of both GPT-3.5 and LLaMA3 8B performance on BioMaze's different biological domain.

## A.12 INTRODUCTION OF PATHWAY

Understanding biological systems is inherently complex due to the numerous interacting molecules, processes, and environmental factors involved. These systems operate with intricate interactions that result in non-linear, multi-layered, and dynamic behaviors. To address this complexity, biological researchers use pathway graphs as structured blueprints to simplify these systems into organized structures that consist of basic interactions. The linear reactions, cyclical relationships, or the local network of pathways offer snapshots of how a system behaves under specific conditions and enable researchers to predict how changes in one molecule or interaction can affect the entire system. Pathway graphs also provide a structured, static representation of dynamic processes, helping researchers understand the sequence of events even as the system changes over time.

## A.13 DETAILED RELATED WORK

**Biological Scientific Question Answering** Previous studies have explored the potential of language models in the biological scientific domain. MEDHOP (Welbl et al., 2018) and PubMedQA (Jin et al., 2019) investigated biological scientific question answering in the form of reading comprehension. BioASQ-QA (Krithara et al., 2023) proposed a realistic question-answering benchmark for the actual information needs of biomedical experts. Beyond textual QA, several works have also studied multimodal scientific ability (Lu et al., 2022). Additionally, other studies have explored biomedical domain tasks (Vilares & Gómez-Rodríguez, 2019; Jin et al., 2021; Pal et al., 2022). Most existing tasks in the biological sciences concentrate on knowledge probing, assessing how well models understand biological information. In contrast, our work, BioMaze, is the first to focus on models' reasoning abilities within the biological scientific domain, specifically targeting phenomena observed in experiments about biological pathways.

A few studies have examined language models' ability to understand biological pathways. Chatpathway (Li et al., 2023b) and Azam et al. (2024) specifically investigated language models' capacity for completing biological pathways. However, these studies mainly focus on probing biological pathway knowledge, i.e., determining whether language models possess the relevant pathway information. In contrast, this work introduces a novel task that employs pathway models for practical biological pathway phenomenon reasoning, bridging the gap between pathway network knowledge and its implications. We compare this work with previous biological datasets in Table 8.

Table 8: Comparison of previous biological scientific question answering tasks and BioMaze.

|  | Domain | Question Form | Task Types |
|---|---|---|---|
| MEDHOP Welbl et al. (2018) | Scientific | Choice | Multi-hop reading comprehension |
| PubMedQA Jin et al. (2019) | Scientific | True/False | Reading comprehension |
| HEAD-QA Vilares & Gómez-Rodríguez (2019) | Healthcare | Choice | Knowledge probing and reasoning |
| MedQA Jin et al. (2021) | Medical | Choice | Reading comprehension |
| MedMCQA Pal et al. (2022) | Medical | Choice | Knowledge probing and reasoning |
| BioASQ-QA Krithara et al. (2023) | Scientific | True/False and Open-Ended | Knowledege probing |
| ChatPathway Li et al. (2023b) | Pathway and biochemical | Open-Ended | Knowledge probing |
| Azam et al. (2024) | Pathway and gene | Choice | Knowledge probing |
| BioMaze (Ours) | Pathway for the scientific | True/False and Open-Ended | Reasoning in biological pathway |

**Graph-augmented Language Model** Several studies have explored augmenting large language models (LLMs) with graph data. In particular, some works enhance LLMs by encoding graph data as text (Ye et al., 2023; Wang et al., 2024; Fatemi et al., 2023), or tuning LLMs specifically for graph-based tasks (Liu et al., 2023a; Tang et al., 2024; He et al., 2024; Zhao et al., 2023; He & Hooi, 2024). By augmenting LLMs with graph data, they have been applied to knowledge-based QA (Sun et al., 2023; He et al., 2024; Li et al., 2023a; Jin et al., 2024; Cheng et al., 2024), and to graph-oriented tasks like graph property prediction (Wang et al., 2024; He et al., 2023). A few other studies leverage graph structures during LLM reasoning to tackle complex tasks (Jiang et al., 2023; Besta et al., 2024).

Unlike tasks in previous works, this study addresses whether reasoning in biological systems can be enhanced by pathway graphs, which act as a *structured blueprint* for reasoning about the system's states. It is not sufficient to simply identify the correct paths in the pathway graph to find the answer. Instead, it is necessary to perform deductive reasoning about the events that occur when the system is intervened upon under specific conditions and to predict the resulting states and mechanisms of the intervened system.

For large graph databases, most works enable LLMs to access graph data through retrieval mechanisms (He et al., 2024; Li et al., 2023a), while a few studies have explored using LLMs as interactive agents (Yao et al., 2023; Shinn et al., 2023; Zhao et al., 2024) to navigate and explore vast graph databases (Sun et al., 2023; Jin et al., 2024). In this work, we introduce an agent-based interactive graph exploration approach using subgraph navigation-based browsing, which is more efficient and offers enhanced navigation capabilities for pathway database.

