# OpenReview forum: "Benchmarking and Enhancing Large Language Models for Biological Pathway Reasoning"
_ICLR.cc/2025/Conference — Submitted to ICLR 2025_

### Official Review · Reviewer_MoHE · 2024-11-03

**Soundness:** 3
**Presentation:** 3
**Contribution:** 3
**Rating:** 6
**Confidence:** 3

**Summary:**

This study explores the under-examined ability of LLMs to reason about biological pathways, particularly focusing on how system perturbations affect downstream biological processes. The authors introduce the BioMaze dataset, a benchmark designed to assess LLMs’ reasoning on how various interventions, like mutations, infections, or treatments, impact downstream targets through complex pathway mechanisms across different biological contexts. With this dataset, the authors then test LLMs with reasoning techniques such as Chain-of-Thought (CoT) and graph-augmented methods, and they find that while LLMs can understand basic biological mechanisms, they struggle with predicting effects after perturbations. To enhance the reasoning ability of LLMs, the authors also developed PathSeeker. In this novel approach, the LLM agent navigates pathway subgraphs to improve performance in pathway reasoning, particularly in scenarios with biological perturbations.

**Strengths:**

1. BioMaze benchmark for biological pathway reasoning: The authors present BioMaze, a benchmark dataset designed to evaluate LLMs’ reasoning abilities within a biological context. BioMaze focuses on assessing how well LLMs comprehend and reason about complex biological pathway phenomena, including cause-effect relationships in natural and perturbed conditions. Curated from the literature, this dataset includes high-quality questions and answers generated with Llama 3.1405B and GPT-4. Covering multiple biology subfields, BioMaze undergoes extensive filtering and validation to ensure relevance, accuracy, and diversity of pathway scenarios.
2. Pathway graph augmentation via PATHSEEKER agent model: Given that biological pathways are naturally structured as networks, the authors incorporate pathway graph data to improve LLM reasoning. They introduce PATHSEEKER, a novel graph-augmented agent that navigates pathway subgraphs to enrich LLM understanding and support reasoning in complex pathway contexts. This approach allows LLMs to access and utilize structural information essential for nuanced pathway reasoning, particularly in scenarios involving biological interventions.
3. Comprehensive evaluation and analysis: The paper conducts a thorough evaluation across multiple LLM models and experimental settings, systematically analyzing LLM performance with and without pathway graph augmentation. Additionally, the ablation study of PATHSEEKER explores its effectiveness by examining API usage, step distribution, and performance impact. These analyses further strengthen the value of pathway augmentation, validating the importance of PATHSEEKER in enhancing LLMs’ reasoning capabilities in biological pathway contexts.

**Weaknesses:**

1. Limited evaluation method for open-ended questions: outputs from different LLMs are evaluated by another LLM, specifically using the Llama 3.1 405B model, which is considerably powerful but would be costly to replicate the results. It would be more helpful if the authors could consider some alternatives, such as using rule-based keyword-matching or for example, using ROUGE score or embedding-based summarization methods to compare how similar or dissimilar answers from LLMs are to the ground truth answers. Another alternative could be to construct different evaluation methods based on the failure modes discovered later from the error analysis study.
2. see questions

**Questions:**

1. Pathway graph limitations: This paper highlights that faulty reasoning persists even with pathway augmentation, especially with perturbations. Could the authors provide more insight into potential sources of error in the pathway graph data? Is it the case that some specific cases or graph structures are more challenging for the LLM to navigate, and some are easier for LLMs to handle?
2. Handling multi-step reasoning decline: Given that CoT reasoning shows decreased accuracy with increased steps, have the authors considered alternative strategies or mechanisms, such as hierarchical reasoning, to mitigate this drop in performance, or are those questions just naturally challenging?
3. Error analysis: The error analysis indicates that omissions remain an issue with PATHSEEKER. What approaches might the authors consider to address these issues, especially when key pathway branches are missed? Could further database expansion, enhanced subgraph search criteria, or developing a different graph search algorithm improve the performance?
4. Using RAG: would authors consider incorporating RAG into this framework given the graph structure of biological pathways? Specifically, RAG could allow the model to retrieve specific or relevant information from related literature or pathway databases. This retrieval would provide the LLM with dynamic access to more detailed and more recent biological knowledge, instead of the graph structure constructed from a fixed database KEGG, as currently used in the paper.
5. Evaluator setting in this paper: this paper proposes using the llama 405B model as the evaluator model for LLM's outputs, as this is costly to run multiple times, would authors consider any alternative evaluation approaches such as applying rule-based methods or using alternative LLMs to strengthen the statistical validity of the benchmarking results?

---

> ### Author Response · Authors · 2024-11-22
>
> ## Open-ended answer evaluation method
>
> Thank you for the insightful question! For the evaluation of open-ended tasks, we
> utilized LLaMA-3.1 405B, which we found to provide evaluation quality comparable to GPT-4. Additionally, we experimented
> with other models, including GPT-3.5 and LLaMA-3.1 70B, and obtained the following results verified by human assessment:
>
> |                               | LLaMA 3.1 405B | GPT 4 | GPT 3.5 | LLaMA 3.1 70B |
> |-------------------------------|----------------|-------|---------|---------------|
> | Acc with human validation (%) | 96             | 96    | 94      | 93            |
>
> The observed inconsistencies between model and human evaluations often arise in cases where the generated answer is
> close to the ground truth but expressed in a general manner or lacks specific details. This highlights a trade-off
> between evaluation precision and computational cost. For instance, GPT-3.5 is more cost-efficient, while GPT-4 offers
> higher accuracy at a greater expense.
>
> Regarding evaluation methods like rule-based approaches, similarity metrics, or ROUGE scores, they are not well-suited
> for open-ended generation tasks. The key challenges include:
>
> 1) Matching Biological Entities: For instance, evaluating answers that equate terms like coatamer protein II complex and
>    COPII requires advanced engineering efforts that rule-based methods struggle to handle.
>
> 2) Different Expressions of the Same Fact: For example, the standard answer might state, "NleF slows the intracellular
>    trafficking of tsVSVG from the endoplasmic reticulum to the Golgi," while the model-generated response suggests, "
>    NleF
>    causes a delay or blockage in the anterograde transport trafficking of tsVSVG, leading to changes in its
>    intracellular
>    localization." While semantically equivalent, such variations are difficult to assess using rule-based, similarity,
>    or
>    ROUGE metrics.
>
> Given the additional costs and challenges of evaluating open-ended tasks, we developed the True/False task in BioMaze to
> address these problems, which offers both greater convenience and accuracy. The True/False questions are designed
> as
> probing forms of biological pathway queries, which, while still challenging as our experiments
> demonstrate, simplify evaluation.
>
> ## Pathway graph limitations (potential sources of error when using the pathway graph data)
>
> Thank you for your suggestion! To enhance the clarity of the error categories and provide
> more detailed insights, we have included representative examples for each error category in Appendix A.2 of the revised
> draft (due to space constraints in the main paper). In particular, we present error cases related to the pathway
> graph-augmented method, as illustrated in Figures 10 and 12.
>
> One source of errors in the pathway-augmented method arises from the inherent complexity of graph data, especially in
> pathways with self-circulatory or multi-branch structures. For example, in Figure 10, the question asks:
> "What is the effect of heparin deficiency on the formation and degradation of Ang II in these peritoneal cell cultures?"
> Here, the model's reasoning process considered the pathway involving the degradation of Ang II but overlooked the more
> critical pathway concerning the conversion of Angiotensin I to Angiotensin II. This omission led to an incorrect
> conclusion.
>
> The challenge arises from the textual representation of pathway graphs. Although we developed a DFS-based graph
> sequentialization method to better capture graph features, sequential LLMs still face difficulties in understanding and
> reasoning about complex graph structures. This limitation is especially pronounced when they need to perform deductive
> reasoning across multiple branches or navigate self-circulatory pathways simultaneously.

---

> ### Author Response · Authors · 2024-11-22
>
> ## Handling multi-step reasoning decline during cot
>
> Thank you for this inspiring idea! To address this, we conducted additional
> experiments by designing a hierarchical reasoning method. This approach requires the LLM to first outline the pathway
> potentially involved in the question as a reasoning plan, and then conduct reasoning based on this self-proposed
> pathway. This method, which we denote as CoT-Self-Pathway, is similar to the pathway-augmented reasoning method, except
> that the pathway is self-generated by the LLM rather than provided by an external database.
>
> Below are the results comparing CoT-Self-Pathway to standard CoT and our PathSeeker on the True/False task:
>
> |           |                  | Inquiry Type |           | Extra Condition |            | Investigation Target |             |          |
> |-----------|------------------|--------------|-----------|-----------------|------------|----------------------|-------------|----------|
> |           |                  | Normal       | Perturbed | Natural         | Intervened | Single               | Interaction | Function |
> | GPT-3.5   | CoT              | 77.03        | 67.13     | 73.65           | 68.92      | 68.92                | 79.26       | 71.85    |
> |           | CoT-Self-Pathway | 79.11        | 67.81     | 75.14           | 66.19      | 69.27                | 81.23       | 71.02    |
> |           | PathSeeker       | 78.85        | 74.44     | 77.63           | 74.36      | 78.01                | 81.66       | 73.78    |
> | LLaMA3 8B | CoT              | 81.77        | 71.63     | 79.04           | 70.67      | 79.73                | 84.35       | 71.52    |
> |           | CoT-Self-Pathway | 79.64        | 72.83     | 81.35           | 70.53      | 81.95                | 82.39       | 68.41    |
> |           | PathSeeker       | 83.08        | 75.84     | 82.14           | 72.27      | 81.07                | 86.62       | 75.01    |
>
> As the results indicate, CoT-Self-Pathway generally achieves performance comparable to standard CoT. We
> observed that the self-generated pathways tend to be more abstract and less detailed or comprehensive compared to
> pathways retrieved from graph databases. In some cases, these self-proposed pathways are more contextually aligned with
> the question, which can be an advantage.
>
> The performance of this method is primarily constrained by the quality of the pathways generated by the LLM itself.
> Since the LLM-generated pathways lack the details and accuracy of those from a dedicated pathway graph database,
> CoT-Self-Pathway typically does not perform as well as pathway-augmented methods such as PathSeeker.
>
> ## Pathway graph searching improvement
> Thank you for your insightful observation! The limitation of pathway graph
> augmentation, particularly omissions in reasoning, can indeed be mitigated through database expansion and enhanced graph
> search algorithms.
>
> Our method, PathSeeker, specifically focuses on improving graph search efficiency by employing
> LLM-guided subgraph navigation, which has resulted in superior performance compared to other graph augmentation methods.
> For example, we conduct an error analysis including LLM-pruning graph BFS method ToG. Below are the error type
> classification results from the open-ended task:
>
> #### Percentage of Errors Across All Data in Open-Ended Tasks
>
> |           |            | unresolved conclusion | incomplete answer | omission in reasoning | faulty in reasoning |
> |-----------|------------|-----------------------|-------------------|-----------------------|---------------------|
> | GPT3.5    | CoT        | 2.2                   | 6.6               | 9.1                   | 9.9                 |
> |           | ToG        | 3.2                   | 4.4               | 18.5                  | 4.6                 |
> |           | PathSeeker | 0.5                   | 6.0               | 10.7                  | 6.1                 |
> | LLaMA3 8B | CoT        | 1.5                   | 5.5               | 8.2                   | 8.3                 |
> |           | ToG        | 1.6                   | 2.9               | 14.3                  | 2.8                 |
> |           | PathSeeker | 0.7                   | 5.4               | 11.1                  | 3.6                 |
>
> These results indicate that as an LLM-pruning graph BFS method, ToG is more prone to errors from omissions in reasoning,
> due to its less efficient graph navigation strategy. As a result, PathSeeker illustrate higher accuracy in the BioMaze
> evaluation. We believe that further advancements in graph search efficiency could enhance pathway graph recall,
> ultimately boosting
> the model's reasoning capacity.
>
> In this work, we primarily utilized the KEGG pathway graph database. Moving forward, we
> plan to incorporate additional pathway databases, such as Reactome, into the project. This expansion will allow us to
> cover a broader range of scenarios and improve the overall robustness of the method.

---

> ### Author Response · Authors · 2024-11-22
>
> ## Using RAG
>
> Thank you for your insightful feedback! Our current graph database already leverages textual content for
> retrieving nodes and edges, which aligns with the concept of "retrieval from a pathway database." A potential
> Retrieval-Augmented Generation (RAG) approach could indeed incorporate textual modalities to provide additional
> contextual information in the retrieved content.
>
> In practice, existing literature or pathway databases are more accessible and widely used, making this a highly
> practical
> solution. Our method, PathSeeker, is adaptable to any database as long as graph navigation—particularly local subgraph
> searching—can be developed based on the database. We believe this capability is critical for efficiently navigating
> graph-structured databases.
>
> On the other hand, using pure graph data provides an idealized framework for formalizing the reasoning process,
> specifically for testing the capacity of LLMs to perform deductive reasoning on pathway graphs.
> Since this work focuses on Pathway Reasoning, we believe a graph-form database offers the most suitable foundation for
> evaluating the biological pathway reasoning capabilities of LLMs.

---

> ### Author Response · Authors · 2024-11-25
>
> We sincerely hope that our response addresses your questions. We remain available to address any further questions you may have.

---

### Official Review · Reviewer_5QNb · 2024-11-04

**Soundness:** 4
**Presentation:** 4
**Contribution:** 3
**Rating:** 6
**Confidence:** 1

**Summary:**

This paper introduces BioMaze, a large-scale benchmark for evaluating large language models' ability to reason about biological pathways. The authors also introduced PATHSEEKER, a new approach to enhance LLMs' performance on these tasks.
They found that while LLMs can understand basic biological mechanisms but LLMs struggle when asked to reason about perturbations or interventions in biological systems.
Through their experiments, they observed that LLMs perform worse on perturbed systems compared to normal conditions.

**Strengths:**

1. The study is very comprehensive. I like the rigorous experimental design that systematically evaluates different aspects of pathway reasoning.
2. It contributed to the field of BIOLOGICAL PATHWAY REASONING by making benchmarks and the problem formulation combining biological pathway reasoning with LLM capabilities.
3. I also found myself enjoy reading the paper and like the well-structured presentation progressing logically from problem motivation to solution.

**Weaknesses:**

1. The author presents error categorization but it doesn't provide detailed analysis of when and why particular types of errors occur. If the authors can provide more analysis of the occurance, it would be nice.
2. The validation of ground truth answers relies heavily on LLMs themselves (LLaMA 3.1-405B and GPT-4). This circular dependency could reinforce existing model biases.

**Questions:**

1. Maybe the author can try to answer the reason why particular types of errors occur in categorization.
2. May using other models to do ground truth answers validation.

---

> ### Author Response · Authors · 2024-11-22
>
> ## Error categorization reason analysis
>
> Thank you for your insightful suggestions! To improve the clarity of the error categories and provide more detailed
> insights, we have added representative examples of each error category in Appendix A.2 of the revised draft (due to
> space constraints in the main paper). While it is challenging to succinctly summarize the reasons behind each type of
> error, we believe these examples will help improve understanding.
>
> Below, we briefly illustrate a few examples, but we strongly recommend referring to Appendix A.2 for a more
> comprehensive analysis.
> 1. **Omission in Reasoning**
>    This error occurs when critical steps in the reasoning process are omitted, leading to an incorrect final answer.
>    **Example:**
>    **Question:** Does BAMBI enhance or inhibit Wnt-promoted cell cycle progression?
>    **Label:** BAMBI increases Wnt-promoted cell cycle progression.
>    **Error Description**:The model's reasoning only identified BAMBI as a target of beta-catenin but failed to account for its interactions
>    with key components of the Wnt signaling pathway, such as LRP6, FZD10, and DVL1. This omission led to an incorrect
>    conclusion.
>
> 2. **Faulty Reasoning**
>    This error occurs when the reasoning path aligns with the question context but contains significant errors in
>    deducing the events or relationships within that pathway.
>    **Example:**
>    **Question:** What is the effect of GogB-deficient Salmonella on tissue damage and colonization in the gut during long-term
>    chronic infections?
>    **Label:** GogB-deficient Salmonella cause extensive tissue damage and enhanced colonization in the gut during
>    long-term chronic infections.
>    **Error Description**:In this case, the model considered the NF-kappaB signaling pathway but failed to deduce that GogB deficiency leads to
>    increased inflammation. This critical oversight resulted in an incorrect conclusion.
>
> 3. **Unresolved Conclusion**
>    This error occurs when the model cannot provide a definitive answer, indicating uncertainty or suggesting that the
>    answer is unknown.
>    **Example:**
>    **Question:** What is the effect of the E2A-Pbx1 fusion protein on the expression of the novel WNT gene, WNT-16, in pre-B
>    ALL cells?
>    **Label:** The E2A-Pbx1 fusion protein activates the expression of WNT-16 in pre-B ALL cells.
>    **Answer:** Further investigation would be needed to determine the exact effect of the E2A-Pbx1 fusion protein on
>    WNT-16 expression in pre-B ALL cells.
>    **Error Description**:In this case, the model is unable to determine the specific effect of the fusion protein on WNT-16 expression.
>
> 4. **Incomplete Answer**
>    This error occurs when the response omits essential details, such as specific effects or key components of the
>    labeled answer.
>    **Example:**
>    **Question:** How is continual efferocytosis enhanced in macrophages?
>    **Label:** Continual efferocytosis is enhanced in macrophages by the metabolism of apoptotic cell-derived arginine
>    and ornithine to putrescine by macrophage arginase 1 (Arg1) and ornithine decarboxylase (ODC).
>    **Answer:** When macrophages engulf apoptotic cells (ACs), they not only clear the ACs but also metabolize the
>    engulfed cargo. This metabolic reprogramming involves the activation of specific metabolic pathways, such as
>    glycolysis and fatty acid oxidation, to generate energy and metabolites necessary for the efficient clearance of
>    ACs.
>    **Error Description**:In this case, the model highlights metabolic reprogramming but fails to mention the critical role of arginine and
>    ornithine metabolism in continual efferocytosis, resulting in an incomplete answer.

---

> ### Author Response · Authors · 2024-11-22
>
> ## Data correctness validation
>
> To ensure question quality, we employ a two-step process. First, we filter questions using an advanced language model (
> LLaMa 3.1-405B) to assess their relevance and clarity. Subsequently, each question undergoes a final quality check by
> human reviewers.
>
> To validate the answer quality, we require the LLM (LLaMa 3.1-405B) to answer the questions **based on the original
> paper content rather than by themselves**. The model is explicitly instructed to respond with **Undetermined** if it
> cannot confidently generate an answer. Each question is tested five times, and only questions that are consistently
> answered correctly (i.e., aligned
> with the intended label) and not marked as Undetermined in any of the trials are retained. This process helps eliminate
> questions with incorrect labels, ambiguous phrasing, or poor structure.
>
> In the final stage, human experts perform an additional quality check to refine the questions further. Approximately 5\%
> of the data is filtered out at this stage, primarily due to issues such as hint leakage in the question, overly complex
> phrasing (e.g., asking for multiple facts), or poorly defined structure. During this stage, human reviewers also verify
> label correctness, ensuring the dataset's overall reliability and usability.
>
> Through this comprehensive validation pipeline—particularly the human review step—we strive to ensure high data quality,
> with a focus on minimizing LLM errors and enhancing the accuracy of ground truth answers.

---

> ### Author Response · Authors · 2024-11-25
>
> We sincerely hope that our response addresses your questions. We remain available to address any further questions you may have.

---

### Official Review · Reviewer_m7QK · 2024-11-05

**Soundness:** 4
**Presentation:** 3
**Contribution:** 2
**Rating:** 5
**Confidence:** 5

**Summary:**

This paper introduces a benchmark to evaluate LLMs' reasoning abilities about biological pathways including perturbed pathways. The benchmark is diverse and covers different biological domains and scenarios.

The authors' evaluations show that while LLMs understand natural mechanisms well, they struggle with intervention scenarios

The authors propose PATHSEEKER, an LLM agent that navigates pathway graphs using subgraph-based exploration. This approach improves reasoning accuracy, including accuracy for intervention scenarios.

Key contributions:

* BioMaze Benchmark
* Evaluation of LLMs on benchmark
* PATHSEEKER Agent, analysis of its performance on benchmark, its failure modes, and ablation study

**Strengths:**

* The benchmark is a solid contribution. The authors did good work in breaking down the benchmark by various categories.
* PATHSEEKER has promise, though I wish it were better motivated and contextualized within related work within systems biology as well as graph reasoning tasks with LLMs as well as graph-based RAG techniques.
* The breakdown of failure modes for LLM reasoning over pathways, particularly in terms of causality, and showing how the graph augmentation helps is useful. Breaking down the reasons for failure with human validation is also a useful contribution and I wish I saw more of that.

**Weaknesses:**

* With PATHSEEKER, I think there is a lack of motivation for explore the pathways via subgraphs other than "Inspired by how humans browse web networks". I don't disagree with this approach per se, but I don't think the authors motivate doing it this way as opposed to, say for example, adding including the whole graph or a big chunk of it into the prompt template. Indeed, as an experiment I pasted an XML file of a MAPK KEGG map into GPT-4's context window and it fits. And if something doesn't fit, context windows will get bigger. I think the authors should motivate the local approach, for example, by citing work that demonstrates failure modes for graph-based reasoning with LLMs, and citing work that shows how local approaches do better.
* I find it concerning that the authors did not include results for a cutting edge model like GPT-4, Claude, PALM 2 and limited tests to GPT-3.5 and Llama-3 8b, neither of which were fined-tuned for performance in this domain. The gap between GPT-3.5 and GPT-4, as an example, on general medical QA performance is quite large. This makes me worry the benchmark might be already saturating on more advanced models. Budget could have been an issue, but could have fine-tuned GPT-3.5 (perhaps on hold-out data from their benchmark), or they could have used their instance of LLaMa-3.1-405B to answer questions as well as evaluate them. Similarly, they could have used other fine-tuned open source models to evaluate.

**Questions:**

"We then apply multiple data filters and validation steps to ensure the correctness, quality, and relevance
to biological pathways. The correctness of each question is validated by checking whether LLMs
can answer it accurately using the original paper content, allowing us to exclude question-label pairs
with errors. Question quality is ensured through several filters, removing questions that are poorly
defined, unpredictable (e.g., asking for specific measurement values), query more than one fact, are
trivial with answers revealed in the question’s context, or are unrelated to biological pathways. After
all the filters, BioMaze contains 1.3k high-quality questions for biological pathways reasoning"
Can you give me more confidence that these questions all have a single right answer that can be answered from the context? To what degree are the manually verified? Filters are great, but where does the buck stop?

A pathway is essentially a knowledgebase. It would be good to connect this work to recent approaches that use knowledgebase graph structure in RAG, such as GraphRAG. Indeed, generally speaking, contextualization within prior work could be stronger.

Biggest question: Why did you not run eval on cutting-edge larger models or larger open-source models like your LLaMa-3.1-405B, or fine-tuned SLMs. Bit sus. Willing to upgrade review if this concern is addressed.

---

> ### Author Response · Authors · 2024-11-22
>
> ## Motivation for using subgraph methods rather than the whole graph
>
> Thank you for your insightful question! While it is true that an individual pathway file,
> such as MAPK, can fit within the context window of an LLM, typical and realistic biological research scenarios often do
> not have pre-given information of which specific pathway is relevant to the question. Furthermore, the reasoning process
> may involve interactions or activations spanning multiple human-splitted pathway maps.
>
> To address this challenge, we combine **all** KEGG pathways into a **single**, comprehensive graph that serves as the
> augmentation database for all the questions in the BioMaze, as we described in "Pathway Graph Database" of Subsection
> 3.3. This combined graph is too large to fit within the LLM’s context window in its entirety. As a result, PathSeeker,
> along with the baselines we selected, focuses on methods capable of dynamically identifying and extracting relevant
> subgraphs from large graph databases. This ensures our approach is both scalable and well-suited to the complexities of
> real-world biological pathway reasoning.
>
> We have modified the description of the graph database in the revised draft to clarify the motivation and methodology
> behind our approach.
>
> ## Evaluation of cutting-edge models as backbone
>
> Thanks for the suggestion of experiment! We conduct experiment with backbone LLaMa 3.1-405B, and here is the result on
> True/False tasks:
>
> |                  | Inquiry Type |           | Extra Condition |            | Investigation Target |             |          |
> |------------------|--------------|-----------|-----------------|------------|----------------------|-------------|----------|
> | LLaMa 3.1 405B   | Normal       | Perturbed | Natural         | Intervened | Single               | Interaction | Function |
> | Viliana (0 Shot) | 84.58        | 74.19     | 81.63           | 76.26      | 80.44                | 85.14       | 76.88    |
> | Viliana (2 Shot) | 85.47        | 73.68     | 82.21           | 75.91      | 83.34                | 83.83       | 76.65    |
> | CoT (0 Shot)     | 87.54        | 78.98     | 85.05           | 80.46      | 86.14                | 87.85       | 74.91    |
> | CoT (2 Shot)     | 86.09        | 78.65     | 83.44           | 79.45      | 85.42                | 88.14       | 77.74    |
> | ToG              | 87.31        | 79.08     | 84.19           | 78.20      | 86.04                | 89.50       | 75.36    |
> | CoK              | 83.85        | 77.51     | 81.91           | 77.42      | 84.42                | 86.42       | 78.31    |
> | G-Retriever      | 86.87        | 79.83     | 84.43           | 80.16      | 86.65                | 89.21       | 78.77    |
> | PathSeeker       | 89.43        | 81.80     | 84.21           | 82.25      | 87.83                | 87.09       | 81.82    |
>
> Here are some key observations:
>
> 1) Cutting-edge model achieve higher performance: Overall, LLaMa 3.1-405B achieves an 8% performance improvement
>    compared to
>    the 8B version.
>
> 2) Persistent gap in intervention scenarios: The results demonstrate that cutting-edge models exhibit varied performance
>    under different settings. A noticeable performance gap remains between natural and perturbed cases.
>    This
>    indicates that even state-of-the-art models struggle with reasoning about interventions in biological pathways
>    compared
>    to their better understanding of natural pathway states.
>
> 3) Effectiveness of PathSeeker: Our proposed method, PathSeeker, still shows improvement comparing to CoT method,
>    particularly in scenarios involving interventions.
>
> We will continue updating this dataset by incorporating data from more recent publications that were not available
> during the model’s pretraining phase, aiming to enhance the evaluation of LLMs in realistic biological research
> scenarios involving pathway reasoning.
>
> ## Further related work
> Thank you for highlighting this relevant area of research! The graph-augmented baselines we explore—such as ToG, CoK,
> and G-Retriever—are indeed closely tied to graph-based retrieval methods, as discussed in Section 2 of the paper. We
> have incorporated additional related works, such as GraphRAG and GraphReader, in the revised draft.

---

> ### Author Response · Authors · 2024-11-22
>
> ## Answer validation during data creating and filtering
> To ensure question quality, we employ a two-step process. First, we filter questions using an advanced language model (
> LLaMa 3.1-405B) to assess their relevance and clarity. Subsequently, each question undergoes a final quality check by
> human reviewers.
>
> To validate the answer quality, we require the LLM (LLaMa 3.1-405B) to answer the questions based on the original
> paper's content. The model is explicitly instructed to respond with **Undetermined** if it cannot confidently generate
> an
> answer. Each question is tested five times, and only questions that are consistently answered correctly (i.e., aligned
> with the intended label) and not marked as Undetermined in any of the trials are retained. This process helps eliminate
> questions with incorrect labels, ambiguous phrasing, or poor structure.
>
> In the final stage, human experts perform an additional quality check to refine the questions further. Approximately 5\%
> of the data is filtered out at this stage, primarily due to issues such as hint leakage in the question, overly complex
> phrasing (e.g., asking for multiple facts), or poorly defined structure. During this stage, human reviewers also verify
> label correctness, ensuring the dataset's overall reliability and usability.

---

> ### Author Response · Authors · 2024-11-25
>
> We sincerely hope that our response addresses your questions. We remain available to address any further questions you may have.

---

> ### Author Response · Authors · 2024-11-27
>
> ## Additional result of evaluating cutting-edge models as backbone
> We further evaluated LLaMA 3.1-405B on BioMaze open-ended tasks. We apologize for the delay, as to minimize potential
> model
> bias during evaluation, we employed GPT-4 as the evaluation LLM for LLaMA 3.1-405B. Our analysis of the evaluator
> demonstrated that GPT-4
> exhibits evaluation accuracy comparable to LLaMA 3.1-405B, with both achieving 96% consistency with human. The
> performance
> results are as follows:
>
> |                | Inquiry Type |           | Extra Condition |            | Investigation Target |             |          |
> |----------------|--------------|-----------|-----------------|------------|----------------------|-------------|----------|
> | LLaMa 3.1 405B | Normal       | Perturbed | Natural         | Intervened | Single               | Interaction | Function |
> | CoT (0 Shot)   | 85.38        | 75.91     | 82.17           | 74.96      | 79.91                | 74.05       | 81.26    |
> | CoT (2 Shot)   | 84.71        | 75.70     | 81.65           | 74.81      | 77.79                | 83.50       | 80.31    |
> | ToG            | 86.43        | 79.25     | 84.75           | 76.78      | 84.39                | 79.09       | 81.37    |
> | CoK            | 84.12        | 73.42     | 79.71           | 74.19      | 80.73                | 70.03       | 77.48    |
> | G-Retriever    | 85.78        | 80.17     | 85.80           | 75.10      | 83.00                | 77.47       | 83.28    |
> | PathSeeker     | 88.24        | 83.82     | 88.20           | 79.97      | 86.54                | 82.31       | 85.76    |
>
> The key observations are similar to the result on True/False tasks:
>
> 1) Cutting-edge model achieve higher performance: Overall, LLaMA 3.1-405B achieves a 5% performance improvement compared
>    to the 8B version.
>
> 2) Persistent gap in intervention scenarios: A noticeable performance gap remains between natural and
>    perturbed/intervened cases. This is one of the key conclusions of our benchmarking. Despite the backbone model's
>    stronger knowledge and reasoning abilities, interventions in biological systems still pose significant challenges for
>    the LLM's reasoning.
>
> 3) Effectiveness of PathSeeker: Our proposed method, PathSeeker, demonstrates improved performance compared to the CoT
>    method, particularly in scenarios involving interventions.

---

### Official Review · Reviewer_TncJ · 2024-11-08

**Soundness:** 3
**Presentation:** 3
**Contribution:** 3
**Rating:** 6
**Confidence:** 3

**Summary:**

This paper addresses a gap in LLMs to reason biological pathways, especially with complex perturbations, interventions, and varying conditions. To address this gap, the authors first introduce a new benchmark, BioMaze, that contains 1.3k high-quality questions for biological pathways reasoning.

Next, the paper evaluates LLMs on BioMaze with existing reasoning methods and finds that they struggle with perturbations. Then the authors propose a new reasoning approach, PathSeeker, that reasons through subgraph-based navigation within pathway graph. PathSeeker achieves better performance in biological reasoning.

**Strengths:**

1. Clear identification of research gap: I think it is an interesting question whether LLMs can reason on biological pathways, and how well they do it. The authors have identified the limitations here clearly.

2. Innovative benchmark: BioMaze is a valuable contribution to the field, providing a systematic evaluation framework for assessing LLM performance across various dimensions of biological pathway reasoning.

**Weaknesses:**

1. Data presentation is not very clear. For example, when the paper evaluates the performance of different models and reasoning methods, it simply writes "performance" without defining the metrics. Therefore, it is not clear whether a higher number means a better performance. In Table 2 and 3, the authors underline the lowest results, which is confusing.

2. Baseline choice is not clear. The paper uses CoT as a baseline in 5.3.1 Task Analysis. I think a better baseline may be a method with pathway graph augmentation since PathSeeker also uses pathway graph augmentation.

3. Analysis is not thorough enough. If the authors want to claim that PathSeeker reduces the performance gap between natural and intervened/perturbed groups, then they should provide more evidence and analysis on them.

**Questions:**

1. In Figure 4, how are the lines fitted? For the right figure (open-ended questions), the gap between CoT and PathSeeker is very small. What is the standard deviation?

2. In Table 2, Table 3, Table 6, and Figure 5, please add what metrics and units are used. Also add evaluation method in Experiment section.

---

> ### Author Response · Authors · 2024-11-22
>
> ## Data (metrics and units) presentation
> Thank you for the helpful feedback. We apologize for any confusion caused by the description of our metric. We
> introduced the metric in Subsection 5.1. For True/False tasks, we compute accuracy averaged across the True and False
> labels to
> account for label imbalance in the dataset (50% for random guessing baseline). For open-ended tasks, the LLM is used to
> evaluate the accuracy of generated answers by comparing them to the ground truth and determining whether they are
> correct or incorrect. In this study, we use the LLaMa 3.1-405B model as the evaluator, with five in-context examples.
> The performance of the evaluator is further analyzed in Appendix A.8. We improved the metric description subsection.
>
> We add the metric description to Table 2, 3, 6 and Figure 5 in the revised draft. Thank you for the feedback on this
> matter.
>
> ## Meaning of lowest results in Tables 2 and 3
> In Tables 2 and 3, higher metric values indicate better performance. In these results, we underline the lowest values in
> each dimension to highlight the more challenging setting. We further explain the meaning of the underline in the revised
> draft.
>
> ## Additional baselines for analysis experiment
> Thank you for your thoughtful suggestion. We chose CoT and PathSeeker as baselines to represent two distinct reasoning
> approaches: independent reasoning by LLMs and reasoning augmented by graph structures.
> As shown in the main experiment in Subsection 5.2, PathSeeker effectively utilizes pathway graphs,
> making it the most representative method for illustrating how graph-augmented reasoning works.
>
> To further explore other graph augmentation methods, we also analyzed the ToG method. Below are the error type
> classification results from the open-ended task:
>
> #### Percentage of Errors Across All Data in Open-Ended Tasks
>
> |           |            | unresolved conclusion | incomplete answer | omission in reasoning | faulty in reasoning |
> |-----------|------------|-----------------------|-------------------|-----------------------|---------------------|
> | GPT3.5    | CoT        | 2.2                   | 6.6               | 9.1                   | 9.9                 |
> |           | ToG        | 3.2                   | 4.4               | 18.5                  | 4.6                 |
> |           | PathSeeker | 0.5                   | 6.0               | 10.7                  | 6.1                 |
> | LLaMA3 8B | CoT        | 1.5                   | 5.5               | 8.2                   | 8.3                 |
> |           | ToG        | 1.6                   | 2.9               | 14.3                  | 2.8                 |
> |           | PathSeeker | 0.7                   | 5.4               | 11.1                  | 3.6                 |
>
> These results indicate that as an LLM-pruning graph BFS method, ToG is more prone to errors from omissions in reasoning,
> likely due to its less efficient graph navigation strategy.
> Interestingly, the phenomenon that ToG with GPT 3.5 as the backbone performed worse than with LLaMA3 8B could be
> attributed to GPT 3.5's shorter context length (4096 tokens) compared to LLaMA3’s (8192 tokens),
> which limits the extent of graph navigation and may exacerbate pathway omissions.
>
> ## Evidence for PathSeeker reducing gap between natural and intervened/perturbed groups
> Thank you for this valuable suggestion! To better demonstrate how PathSeeker enhances intervention reasoning,
> we present a comparison of the performance gap between natural and intervened/perturbed groups below:
>
> #### True/False Task: Natural - Intervened/Perturbed (Lower is Better)
>
> |           |              | Inquiry Type Gap | Extra Condition Gap |
> |-----------|--------------|------------------|---------------------|
> | GPT 3.5   | CoT (2 Shot) | 9.89             | 4.73                |
> |           | PathSeeker   | 4.41             | 3.27                |
> | LLaMa3 8B | CoT (2 Shot) | 10.14            | 8.36                |
> |           | PathSeeker   | 7.24             | 9.87                |
>
> #### Open-ended Task: Natural - Intervened/Perturbed (Lower is Better)
>
> |           |              | Inquiry Type Gap | Extra Condition Gap |
> |-----------|--------------|------------------|---------------------|
> | GPT 3.5   | CoT (2 Shot) | 9.02             | 6.97                |
> |           | PathSeeker   | 9.93             | 4.79                |
> | LLaMa3 8B | CoT (2 Shot) | 12.64            | 8.54                |
> |           | PathSeeker   | 10.42            | 6.24                |
>
> The results indicate that PathSeeker achieves smaller performance gaps between natural and intervened/perturbed groups
> compared to CoT in most scenarios.
> This suggests that leveraging pathway graphs improves reasoning for intervention cases.

---

> ### Author Response · Authors · 2024-11-22
>
> ## Figure 4's fitting method and standard deviation
> We fitted the lines using a third-order polynomial curve with NumPy's polyfit method.
>
> Thank you for suggesting the inclusion of standard deviation! In the revised draft, we have incorporated the standard
> error, calculated from five independent test runs, into Figure 4. The results show that the performance gap is
> significant compared to the standard deviation, particularly for questions requiring a larger number of reasoning steps.
>
> The observed phenomenon where "the gap between CoT and PathSeeker is very small" predominantly occurs for questions
> involving fewer reasoning steps. For questions requiring more reasoning steps, however, the gap becomes more pronounced.

---

> ### Author Response · Authors · 2024-11-25
>
> We sincerely hope that our response addresses your questions. We remain available to address any further questions you may have.

---

### Author Response · Authors · 2024-11-22
**Draft Modification Summary**

We thank all the reviewers for their insightful suggestions and questions regarding the paper. Below, we summarize the
modifications made to the paper for clarity:

1. **Metric Descriptions**
   As suggested by reviewer TncJ, we have added detailed descriptions of the metrics used in Tables 2, 3, and 6, as well
   as Figure 5. Additionally, we have clarified the meaning of the lowest results presented.

2. **Standard Deviation in Figure 4**
   Based on reviewer TncJ's suggestion, we have included the standard deviations in Figure 4.

3. **Description of Pathway Graph Database**
   Following reviewer m7QK's feedback, we have provided a more detailed explanation of the pathway graph database. This
   database consists of a single, large graph created by merging all KEGG pathway maps and pathways.

4. **Related Work**
   As recommended by reviewer m7QK, we have expanded the Related Work section to include discussions of GraphRAG and
   GraphReader.

5. **Failure Reason Cases**
   In response to the queries from reviewers 5QNb and MoHE, we have added case examples for each type of failure reason
   in Appendix A.2. These examples aim to provide a better understanding of how errors occur during the reasoning
   process.

6. **Other Modification in Representation**
   We revised some of the paper's presentation to enhance clarity, such as the illustration in Table 1.

For the convenience of review, we have highlighted the modifications in blue.

---

### Author Response · Authors · 2024-11-25
**A Kind Reminder for Reading the Response**

Thank you once again for your thoughtful questions and suggestions! We sincerely hope our response and revisions have addressed your concerns. As the rebuttal period is closing soon, we kindly ask if you could read our response to ensure it effectively mitigates your concerns. We would greatly appreciate your feedback and remain available to address any additional questions or concerns.

Thank you,

The Authors

---

### Author Response · Authors · 2024-11-29
**A Kind Reminder for Reading the Response**

We sincerely thank all the reviewers once again for their time and effort in providing insightful questions and suggestions regarding our work. We have made every effort to address these questions and incorporate the suggestions into our revisions. We sincerely hope our responses and updates adequately address your concerns.

We kindly request that you review our responses to ensure they effectively address your feedback. We would greatly appreciate any additional comments or suggestions and remain available to further discuss or address any remaining questions or concerns.

Thank you once again for your valuable feedback and guidance.

Sincerely,

The Authors

---

### Author Response · Authors · 2024-12-01
**A Kind Reminder for Reading the Response**

As the rebuttal period is nearing its end, we kindly request the reviewers to review our response to ensure it effectively addresses your concerns. We would greatly value your feedback and are available to clarify or address any additional questions you may have.

Thank you for your time and consideration.

Sincerely,

The Authors

---

### Author Response · Authors · 2024-12-03
**A Kind Reminder for Any Feedback**

As the discussion period comes to a close, we kindly request that reviewers provide any feedback they have on our
response and remain available to address
any additional questions or concerns.

Thank you,

The Authors

---

### Author Response · Authors · 2024-12-04
**Summary of Questions and Responses**

We thank the reviewers for all their valuable questions and suggestions. We sincerely hope our response and revisions
have addressed their concerns.

For clarity, we summarize all questions raised by reviewers and our corresponding responses here:

### Additional Experiments to Address Specific Questions:

1. **Evaluation of Cutting-Edge Models as Backbone**
   We included experimental results using larger language model backbone (LLaMA 3.1 405B) on BioMaze.

2. **Additional Baselines for Analysis**
   We analyzed additional baselines, including ToG, and explained why we use PathSeeker as the representative
   graph-augmented method during analysis.

3. **Handling Multi-Step Reasoning Decline in Chain-of-Thought (CoT) Approaches**
   We evaluated the hierarchical CoT reasoning approach as proposed by the reviewer.

### Additional Analysis:

1. **Evidence for PathSeeker Reducing the Gap Between Natural and Perturbed Groups**
   We illustrated the gap between natural and intervened/perturbed groups to provide evidence of PathSeeker's
   effectiveness.

2. **Error Categorization and Analysis**
   We added error cases and detailed analyses for each category in Appendix A.2 to enhance understanding.

### Clarification of Confusion and Misunderstandings:

1. **Motivation for Using Subgraph Methods Instead of Whole Graphs**
   We employed subgraph-based methods because all the KEGG pathway graphs are merged together as database, which cannot
   be
   processed as a single context input.

2. **Answer Validation During Data Creation and Filtering**
   We explained the ground truth validation process. During data creation, LLaMA 3.1 (405B) was explicitly instructed to
   verify answers against the original paper's content five times. Only questions consistently answered correctly were
   retained, followed by a final expert review to ensure data quality.

### Discussion of Open Questions:

1. **Evaluation of Open-Ended Answers**
   We discussed the challenges of evaluating open-ended answers using rule-based approaches, similarity metrics, or
   ROUGE scores. We also discussed the feasibility of using more cost-effective models, such as GPT-3.5.

2. **Limitations of Pathway Graph Data**
   We added error cases and analyses in Appendix A.2, and discussed challenges such as self-circulatory or multi-branch
   structures in pathway graphs.

3. **Improvement in Pathway Graph Searching**
   We discussed the potential of further improvement in pathway graph searching methods.

4. **Integration of RAG (Retrieval-Augmented Generation)**
   We discussed how to combine RAG into our method, and the meaning of it.

### Paper Modifications:

1. **Data Presentation (Metrics and Units)**
2. **Interpretation of Lowest Results in Tables 2 and 3**
3. **Figure 4's fitting method and standard deviation**
4. **Further related work**
5. **Error cases and analysis added in Appendix A.2**

For details, please refer to the **Draft Modification Summary** in the Response.

---

### Meta-Review · Area_Chair_H5sk · 2024-12-20

**Metareview:**

The paper focuses on using LLMs to reason on biological pathways, specifically considering natural or synthetic interventions.

An important contribution is the BioMaze benchmark, compiling questions/answers based on pathways from the literature.

A first finding is that the considered LLMs perform better when considering un-perturbed pathways than when considering interventions (L2 level as opposed to L1 level, w.r.t. the causality ladder). I am not sure this finding is surprising (L2 is notoriously more difficult to reason with and the data is less wealthy).

The proposed PathSeeker module leverages subgraphs information (as the overall graph integrating e.g. all Kegg pathway graphs is too large),

**Additional Comments On Reviewer Discussion:**

Some issues are not adequately addressed, e.g.,  the quality of the BioMaze benchmark (circular assessment, after Rev. 5QNb) or the comparison with RAG approaches (although the authors argue that the use of pathways is the most natural one for the domain).

The area chair encourages the authors to pursue on this very promising line of research.

---

### Decision · Program_Chairs · 2025-01-22

Reject